

# The modelled liquid water balance of the Greenland Ice Sheet

Christian R. Steger[1], Carleen H. Reijmer[1], and Michiel R. van den Broeke[1]

[1]Institute for Marine and Atmospheric Research Utrecht (IMAU), Utrecht University, Utrecht, the Netherlands

*Correspondence to:* Christian R. Steger (C.R.Steger@uu.nl)

**Abstract.** Recent studies indicate that the surface mass balance will dominate the Greenland Ice Sheet's (GrIS) contribution to 21[st] century sea level rise. Consequently, it is crucial to understand the liquid water balance (LWB) of the ice sheet and its response to increasing surface melt. We therefore analyse a firn simulation conducted with SNOWPACK for the GrIS and over the period 1960–2014 with a special focus on the LWB and refreezing. An indirect evaluation of the simulated refreezing

climate with GRACE and firn temperature observations indicate a good model performance. Results of the LWB analysis reveal a spatially uniform increase in surface melt during 1990–2014. As a response, refreezing and runoff also indicate positive trends for this period, where refreezing increases with only half the rate of runoff, which implies that the majority of the additional liquid input runs off the ice sheet. However, this pattern is spatially variable as e.g. in the southeastern part of the GrIS, most of the additional liquid input is buffered in the firn layer due to relatively high snowfall rates. The increase in modelled refreezing

leads to a general decrease in firn air content and to a substantial increase in near-surface firn temperature in some regions. On the western side of the ice sheet, modelled firn temperature increases are highest in the lower accumulation zone and are primarily caused by the exceptional melt season of 2012. On the eastern side, simulated firn temperature increases more gradually and with an associated upward migration of firn aquifers.

## 1 Introduction

The mass balance (MB) of the Greenland Ice Sheet (GrIS) has been negative since the early 1990s (Van den Broeke et al., 2016). Besides increased ice discharge through the acceleration of marine-terminating outlet glaciers, the ice sheet is losing mass through increased surface melt and associated meltwater runoff. The latter process has recently become the dominant contributor to mass loss from the ice sheet (Enderlin et al., 2014). The increase in meltwater runoff and associated decrease of the surface mass balance (SMB) is attributed to processes on various spatial and temporal scales, e.g. the polar amplification

(Bekryaev et al., 2010), the darkening of the GrIS (Tedesco et al., 2016), and is further promoted by the hypsometry of the ice sheet (Mikkelsen et al., 2016; Van As et al., 2017). An accurate quantification of the liquid water balance (LWB) of the ice sheet is important, as it determines how much of the liquid input at the surface ultimately reaches the ocean and contributes to sea level rise. A key parameter of the LWB is meltwater storage in the firn (Rennermalm et al., 2013a) by refreezing and liquid water retention. Previous studies suggest that refreezing strongly depends on the model formulation (Reijmer et al.,

2012; Steger et al., 2017) and that it exhibits the largest inter-model variation of all SMB components (Vernon et al., 2013). Besides the instantaneous effect of retaining liquid water, refreezing also co-determines the future potential of firn to absorb



melt, as it reduces the porosity of the firn (Noël et al., 2017) and releases large amounts of latent heat (Humphrey et al., 2012; Cox et al., 2015), which decreases the firn's cold content.

The hydrology of the GrIS is a complex system, which involves various ill-constrained processes (Fig. 1). At the surface, liquid input is determined by rainfall, evaporation/condensation and melt. In areas where the ice sheet is covered by snow and/or firn,

liquid water is able to percolate vertically. These snow/firn layers may act as a buffer for runoff if liquid water either refreezes (Harper et al., 2012) or remains in its liquid state in firn aquifers (Forster et al., 2014). Such aquifers typically form at locations with relatively high amounts of snow accumulation (Kuipers Munneke et al., 2014) and are thus particularly abundant along the southeastern and northwestern margins of the ice sheet (Forster et al., 2014). A recent study (Poinar et al., 2017) revealed that some aquifers likely drain into crevasses. To what degree the water refreezes there or reaches the bed of the ice sheet

remains largely unknown. Along the southwestern and northeastern margins of the ice sheet, firn aquifers are less abundant. In these areas, percolating water typically refreezes in the firn or runs off over the ice surface. A study by Machguth et al. (2016) suggests that horizontal ice layers could inhibit vertical percolation and render underlying pore space inaccessible for liquid water. The water would hence be forced to flow laterally above such obstacles - either as surface runoff or within the firn.

In the bare ice zone, hydrological processes are better understood: Liquid water flows along surface rivers and may accumulate

in supra-glacial lakes (Arnold et al., 2014) or enter the subglacial system via moulins or crevasses. The amount of water stored in supra-glacial lakes is thereby rather small compared to the magnitude of supra-glacial river fluxes, which drain liquid water efficiently from the surface (Smith 2015). Liquid water flowing into moulins or crevasses enters the en- and subglacial (Lewis and Smith, 2009; Lindbäck et al., 2015) hydrological system of the ice sheet. Here, water may refreeze, accumulate in subglacial lakes or flow along channels to the margins of the ice sheet. The relevance of en- and subglacial water storage is

currently rather uncertain. Rennermalm et al. (2013b) suggests that for a watershed in southwestern Greenland, up to 54 % of meltwater may be retained during one season. It seems however possible that this residual is partly caused by uncertainties in e.g. watershed delineation (Rennermalm et al., 2013b) and inter-basin piracy (Lindbäck et al., 2015). A more recent study for a similar catchment yielded little evidence for meltwater storage in en- and subglacial environments (Van As et al., 2017). In summary, the hydrology of the GrIS represents a complex system of pathways that transport meltwater form the surface of the

ice sheet to the ocean (Chu, 2014).

In this study, we quantify the components of the LWB from the GrIS surface to the firn–ice–transition, using a state-of-the-art snow/firn model. The upper boundary conditions for the model are provided by the regional atmospheric climate model RACMO2.3 (Noël et al., 2015). Potential en- and subglacial liquid water retention is not considered as we only model the upper part of the ice sheet. The primary goal is to quantify the spatial magnitude of the different LWB components and assess

how these mass fluxes evolved over the last decades. Additionally, we evaluate the spatial and seasonal occurrence of refreezing and the impact of this process on the firn structure. Furthermore, we analyse how the horizontal extent of firn aquifers, which act as perennial storage for liquid water, evolves with time. The following section provides a brief description of the model and the observational data used in this study. Subsequently, we assess the performance of the model by comparing its output to remote sensing data (GRACE) and in situ measurements (firn temperatures). Section 4 contains an evaluation of the LWB and

a more detailed analysis of refreezing, runoff and changes in the firn structure.



## 2    Definitions, model and data

### 2.1    Definitions

In this study, we investigate the LWB of the upper part of the ice sheet, namely the snow/firn layer. This layer ranges from the surface down to the firn–ice–transition. If percolating water reaches the bottom of this domain, it is considered to leave the ice sheet as runoff. Potential en- and subglacial storage of liquid water are thus not accounted for. Lateral routing of runoff is currently also neglected. However, this is likely a less relevant issue if only near-surface lateral flow is considered, as surface melt typically reaches higher elevated areas later in the season, which means that lower areas are already depleted of pore space and/or cold content and thus do not provide any more storage volume for upstream runoff. The LWB of the firn layer is defined as

$$\frac{dM_{ret}}{dt} = RA - EV + ME - RF - RU, \tag{1}$$

where $M_{ret}$ is the retained liquid mass, $RA$, $EV$ and $ME$ are surface mass fluxes of rainfall, evaporation and meltwater respectively, $RF$ is internal refreezing and $RU$ is runoff at the bottom of the model domain. In this study, the term evaporation refers to phase changes of water from liquid to gaseous (evaporation) and vice versa (condensation). Formally, the SMB used here equals the climatic mass balance (Cogley et al., 2011), i.e. it includes subsurface processes of liquid water retention and refreezing. The Greenland mass balance (MB) derived to validate the modelled SMB with GRACE data is defined as

$$MB = SMB_{GrIS} + SMB_{PIC} - D + \frac{dM_{ts}}{dt}, \tag{2}$$

where $SMB_{GrIS}$ and $SMB_{PIC}$ are the SMB of the GrIS and the peripheral ice caps/glaciers simulated by SNOWPACK, $D$ is ice discharge across the grounding line from marine-terminating glaciers and $M_{ts}$ is the tundra snow mass.

### 2.1.1    Model data

Snow/firn on the GrIS and the peripheral ice caps/glaciers is modelled with SNOWPACK, a state-of-the-art snow model. SNOWPACK was recently applied in different studies (Groot Zwaaftink et al., 2013; Van Tricht et al., 2016; Steger et al., 2017) to simulate snow and firn in polar regions. The model contains an overburden-dependent densification scheme and simulates the evolution of different microstructural snow properties, which are linked to thermal and mechanical snow quantities (Bartelt and Lehning, 2002; Lehning et al., 2002b, a). We run SNOWPACK on an 11 km horizontal grid and with the same ice mask (Fig. 2) as used in the regional atmospheric climate model RACMO2.3 (Noël et al., 2015). At the snow–atmosphere interface, SNOWPACK is forced with mass fluxes (precipitation, evaporation/sublimation, snow drift and surface melt) from RACMO2.3. Liquid water percolation is simulated with a bucket scheme and the irreducible water content follows the formulation of Coléou and Lesaffre (1998). Fresh snow density is prescribed with an empirical parameterisation that depends on mean annual surface temperature (Kuipers Munneke et al., 2015). The enhanced near-surface snow compaction due to strong winds, which is implemented in SNOWPACK for Antarctic simulations (Groot Zwaaftink et al., 2013), is switched off, because the applied



fresh snow densification scheme already accounts for this effect. A more detailed description of the model setup and the spin-up procedure can be found in Steger et al. (2017).

### 2.1.2 Observational data

To derive a MB for Greenland, we use ice discharge data from Enderlin et al. (2014). This dataset contains annual estimates
of ice discharge from 178 marine-terminating glaciers wider than 1 km. Following Van den Broeke et al. (2016), we neglect seasonal variations in ice discharge and assume that all intra-annual variation in the MB is induced by components of the SMB or by tundra snow. The GRACE gravity field solution for Greenland (Groh and Horwath, 2016) we apply is based on the monthly GRACE solution ITSG-Grace2016 (Mayer-Gürr et al., 2016).

Furthermore, we use firn temperatures that were recorded along a 2700 km transect in northwest Greenland (Fig. 2), referred to
as the NW GrIS transect, to evaluate our simulation. Shallow borehole temperature measurements were conducted at 14 sites between 1952–1955 (Benson, 1962) and repeated in 2013 (Polashenski et al., 2014). The former measurements were taken at a range of 3 to 16.75 m depth (predominantly at 8 m) and were corrected for seasonal influences to obtain an intercomparable, mean annual 10 m temperature. The measurements in 2013 were recorded at a depth between 5–12 m (mainly at 8.5–12 m) and were corrected with the same methodology (Polashenski et al., 2014).

## 3 Model evaluations

The performance of SNOWPACK for the GrIS in terms of snow/firn compaction and firn aquifer extent was already assessed in a previous paper (Steger et al., 2017). Here we use additional observations to evaluate the ability of SNOWPACK forced by RACMO2.3 to simulate the refreezing climate of the GrIS. Because no direct observations of refreezing are available, the evaluation is performed indirectly over the SMB of Greenland and the temperature evolution of the firn.

### 3.1 Model evaluation using GRACE

Due to the large footprint of GRACE, the signal also contains mass variations from Greenland's peripheral ice caps and glaciers and from tundra hydrology; primarily from seasonal snow cover. These signals are thus included in Greenland's MB as explained in Sect. 2.1. Tundra snow cover is not simulated by SNOWPACK but the signal is taken from RACMO2.3 output. In RACMO2.3, seasonal snow is simulated with a single-layer model that does not allow for refreezing and liquid water
retention in the snow (Van den Broeke et al., 2016). All surface melt is hence immediately transferred to runoff. A comparison between the derived cumulative MB and GRACE is provided in Fig. 3a. The cumulative MB and GRACE indicate an excellent agreement both in terms of correlation coefficient and linear trends.

The detrended mean seasonal cycle (Fig. 3b) indicates a good agreement in winter and spring, when changes in cumulative SMB are mainly caused by accumulation of solid precipitation on the glaciated area and the tundra. From May on, the derived
MB shows an earlier and steeper decrease compared to the GRACE signal. The minimum in MB occurs both earlier and with a higher magnitude than in GRACE. These findings are consistent with earlier studies (Van Angelen et al., 2014; Alexander et al.,



2016), in which the average seasonal cycle of the MB and GRACE were compared. A likely contributor to this mismatch is the neglect of the time it takes meltwater runoff to reach the ocean (Van Angelen et al., 2014). On a GrIS-wide scale, Van Angelen et al. (2014) demonstrated that delaying runoff by 18 days could minimize the monthly error between MB and GRACE. A study for a catchment in southwestern Greenland revealed that transit times up to 10 days are required to align the modelled

surface runoff and observed river hydrograph optimally (Van As et al., 2017).

Another uncertainty arises from modelled tundra snow cover and tundra hydrology. In comparison to the ice sheet, snow deposition on the tundra accounts for roughly a third of the modelled mean seasonal amplitude (Fig. 3b). A too early snow ablation in the tundra could hence also contribute to the bias between MB and GRACE. This assumption is supported by a comparison of the simulated snow cover fraction (SCF) with MODIS/Terra Snow Cover data (Hall and Riggs, 2016), which

revealed a too early decrease in modelled SCF in most basins (not shown). Potential causes for this bias are the neglect of refreezing and liquid water retention in the relatively simple RACMO2.3 snow model and the poor representation of tundra topography at a horizontal resolution of 11 km. Additionally, heterogeneous snow distribution on a subgrid scale could also contribute to the bias (Aas et al., 2017). Finally, runoff may also be retained in the hydrological system of the tundra; e.g. by temporarily refreezing in soil, ponding on frozen ground (Johansson et al., 2015) and accumulate in surface lakes (Mielko and

Woo, 2006) or terrestrial aquifers. All these processes are currently not represented in our model framework.

### 3.2   Model evaluation with firn temperature measurements

ERA40 reanalysis data, which forces SNOWPACK via RACMO2.3, is available from 1958 onwards. SNOWPACK output for the years 1952–1955, when the first firn temperature dataset along the NW GrIS transect was collected, is thus not available. However, it seems reasonable to assume only small changes in firn temperature between 1952–1955 and the start of the SNOW-

PACK simulation. We therefore compare the 1952–1955 observations to modelled firn temperatures from 1960. From Fig. 4, it appears that SNOWPACK forced by RACMO2.3 slightly overestimates firn temperatures in the higher part of the transect for both periods. For the first period, this bias may be partly caused by the spin-up procedure of the model, where the model is looped over the reference period (1960–1979) to generate the initial firn profile. This means that temperature evolutions before this reference period are not considered. The bias for the second period is more difficult to explain in the absence of continuous

firn temperature measurements. Obviously, the system has a strong memory so these discrepancies could be a result of past biases in the modelled climate (e.g. accumulation and skin temperature).

Between locations B 2-175 and B 2-070, there is a ∼1.6–2.7° C warming in the observations between 1952–1955 and 2013, likely caused by latent heat release due to refreezing. This temperature increase is larger than the modelled, spatially rather uniform warming of ∼0.5° C, which is comparable to the warming in skin temperature over the same period. Possible expla-

nations for this bias are the underestimation of meltwater production at the surface or a too shallow refreezing depth, which enables the released heat to be conducted upwards to the surface and escape to the atmosphere through emission of longwave radiation. In SNOWPACK, percolating water is not allowed to pass unhindered through layers with refreezing capacity, where in reality, liquid water may move to greater depth through piping (Humphrey et al., 2012; Marchenko et al., 2017). At site B 1-010, SNOWPACK simulates a local maximum in firn warming, in agreement with observations. Here, RACMO2.3 simulates



a doubling of the liquid water input between the two periods considered. However, the magnitude of warming in SNOWPACK is somewhat smaller (4.1° vs. 5.7° C), which may again be linked to the neglect of piping.

Other firn temperature records for the GrIS are available, e.g. for the western percolation zone (Humphrey et al., 2012; Charalampidis et al., 2016). These observations also show a substantial warming in the upper ∼10 m firn caused by latent heat release from refreezing. Firn simulations by the IMAU-FDM (Kuipers Munneke et al., 2015) and SNOWPACK (Steger et al., 2017), forced by RACMO2.3, do not reproduce the strong warming observed at these locations, because they tend to overestimate the extent of the bare ice zone on the western GrIS, i.e. the models are incapable of simulating the subsurface warming due to a deficiency of pore space for refreezing. Due to a different densification scheme, which seems to be more accurate for relatively warm conditions (Steger et al., 2017), the overestimation of the bare ice zone is somewhat less pronounced in SNOWPACK than in the IMAU-FDM (Steger et al., 2017). At higher elevations in western Greenland, SNOWPACK does simulate a pronounced warming of the firn layer but there are no in-situ observations available to constrain the magnitude of these changes.

Near-surface snow/firn density depends on external meteorological parameters (e.g. snowfall rates, amounts of surface melt, fresh snow density) and on internal processes (compaction and refreezing rates). One parameter that is particularly uncertain is the fresh snow density. In our study, this quantity is obtained from an empirical relation, which depends on mean annual surface temperature (Kuipers Munneke et al., 2015). The relation was derived with samples from the dry snow zone and is subsequently extrapolated to lower elevation on the ice sheet. Snow/firn density profiles from a transect on the western GrIS (Harper et al., 2012) allow a comparison between observed and modelled near-surface densities: Averaging over the upper 50 cm and all samples yields a value of ∼345 kg m$^{-3}$ for April (i.e. before the onset of seasonal surface melt). For these locations, our fresh snow density parameterisation returns a mean density of ∼405 kg m$^{-3}$. Apparently, the parameterisation overestimates fresh snow density for this region; particularly when considering the fact that measured values represent integrated quantities over the upper 50 cm and have thus already experienced compaction due to overburden pressure. A brief comparison of our fresh snow density parameterisation with near-surface snow density samples obtained on the northern GrIS and for spring (Koenig et al., 2016) supports the assumption that the applied parameterisation yields too high densities for relatively warm climatic conditions. To test SNOWPACK's sensitivity to initial snow densities, an experiment with a lower, spatially uniform fresh snow density of 320 kg m$^{-3}$ was carried out for the western GrIS transect. The selected initial density is comparable to what the recently published parameterisation of Langen et al. (2017) yields for this transect. With this model setting, the mismatch between the observed and modelled bare ice zone extent (and thus the firn warning) was reduced for this specific region. An improved fresh snow density parameterisation seems therefore essential to address this inaccuracy.

## 4 Climatology of the liquid water balance

The evaluation of mass changes and firn temperatures with observations presented in the previous sections inspires sufficient confidence to use SNOWPACK firn data for a description of the LWB of the GrIS. First, we discuss the mean fields and temporal evolution of the LWB components during the simulation period (1960–2014). Subsequently, refreezing, one of the



key components of the balance, and its dependency and influence on firn structure is discussed in more detail. Finally, we analyse the temporal evolution of firn aquifer extent and the partitioning of runoff from ice and snow/firn.

## 4.1 The liquid water balance

Figure 5 shows the temporally averaged (1960–2014) LWB components for the GrIS and the peripheral ice caps and glaciers. Mean fluxes of rainfall and evaporation are typically at least one order of magnitude smaller than melt, runoff and refreezing. Changes in the retained liquid mass ($dM_{ret}/dt$) are even smaller and not presented. Rainfall rates are particularly significant along the southern margin of the ice sheet and in the western ablation zone. For the northeastern part of the GrIS, the contribution of rainfall to the LWB is small and liquid water input at the surface is dominated by melt. The highest melt rates on the GrIS occur along the western ablation zone with a maximum of 129 Gt a$^{-1}$ for Basin 6. The mean spatial runoff pattern is comparable to the one of melt but attenuated by the buffering effect of refreezing. Runoff also peaks in Basin 6 with a mass flux of 85 Gt a$^{-1}$, which accounts for a third of the total GrIS runoff. Averaged over the entire ice sheet, SNOWPACK simulates that almost half (47 %) of the liquid water input at the surface refreezes in snow or firn. This fraction has a high spatial variability and is relatively low for the northeastern basins and for Basin 6, where precipitation is low and bare ice extent relatively large. As a result, refreezing rates in these regions peak more inland in the lower accumulation zone just above the equilibrium line. Refreezing in the ablation zone is, in terms of absolute liquid water retention, only relevant on intra-annual scales. The highest overall refreezing fractions, up to 75 %, are modelled along the wet southeastern margin of the ice sheet (Basin 4).

Time series of the four most relevant LWB components for the eight basins show no distinctive trends for the first half of the simulation period (1960–1989), but do exhibit large interannual variability, particularly for surface melt (Fig. 6). For the second half (1990–2014) however, there is a statistically significant increase in melt in all basins (Table 1). Rainfall, as a further contributor to liquid input, does not exhibit a significant trend for the majority of the basins. Remarkably, the northwestern Basin 8 is the only region with a significant positive trend in rainfall. For all basins, melt rates peak in 2012 when the GrIS experienced unprecedented surface melt both in spatial extent (Nghiem et al., 2012) and magnitude. The exposure of relatively high-elevated regions with cold and porous firn to surface melt is the main reason that refreezing also peaks in all basins during this year. In response to the positive trends in melt, runoff also exhibits a significant increase in all basins between 1990 and 2014 (Table 1). The southwestern Basins 5 and 6 show the strongest increase in runoff per area, which results from the comparably high increase in melt of bare ice. In terms of refreezing and refreezing fraction the response of the basins to increased surface melt is spatially less uniform: The majority of the basins does not indicate a significant trend in refreezing. This means that e.g. for Basins 1 and 2, most of the additional melt is not absorbed in the firn but is running off from the ice sheet, similar to what happens to northern ice caps not connected to the main ice sheet (Noël et al., 2017). Basin 4, which has the highest overall mean refreezing fraction (75 %), is an exception. Refreezing in this basin shows a distinctive positive trend that even exceeds the one in runoff. This is linked to the high amounts of solid precipitation in this basin, which provide enough pore space to absorb the increase in surface melt. Refreezing is also significantly increasing in the northwestern Basins 7 and 8 but with a lower trend than runoff. Significant trends in the refreezing fraction are only apparent in Basin 1 and particularly in Basin 8, where the fraction decreases by ~16 % in 25 years. For the entire GrIS, melt, runoff and refreezing indicate signif-



icant positive trends between 1990 and 2014. The increase in runoff is roughly twice the one in refreezing, which leads to a significant decrease in the GrIS-integrated refreezing fraction of ∼9 % over the 25 years (Table 1). The different response of the eight basins to increasing surface melt is related to refreezing, which in turn is linked to the firn-structure, i.e. the porosity and temperature. This will be discussed in more detail in the following section.

## 4.2 Refreezing and latent heat release

Refreezing is a process that strongly depends on local climate, i.e. accumulation and melt, and therewith on seasonality and elevation (Fig. 7). At the beginning of the melt season, refreezing primarily occurs in the lower parts of the ice sheet, where the melt onset is earliest and meltwater percolates into the cold winter snow layer. For Basin 3–7, low-level refreezing peaks in (late) May while for the northern Basins 1, 2 and 8, the maximum occurs in mid-June. During the course of the melt season, the lower regions are gradually depleted of spore space or cold content and the area of peak refreezing moves upward. For the majority of the basins (e.g., Basin 1, 2 and 7), the availability of pore space is the limiting factor for refreezing in the ablation area. Particularly for Basin 4 and 5 however, this is not the case (Fig. 7). Therefore, refreezing at lower elevations persists throughout the melt season until September. The seasonal decrease in refreezing is caused by the cold content of the firn that gradually decreases. However, even if the entire firn column has become temperate, refreezing persist due to the diurnal temperature cycle, which periodically refreshes the near-surface cold content during night. This underlines the importance of using atmospheric forcing data that resolve variations on subdaily time scales. Peak refreezing rates pass the equilibrium line altitude in June (western Basins 6–8) or in July (northern Basins 1 and 2). For Basins 3–5, it is not possible to define a mean equilibrium line altitude, because these basins have a very narrow ablation zone and at 11 km resolution, many model grid cells close to sea level have a positive SMB due to high accumulation rates. In July, peak refreezing moves beyond the runoff line in all basins. Especially Basins 4 and 5 reveal a percolation zone that stretches over a relatively large vertical extent, with substantial refreezing as high as 500–750 m above the modelled runoff line. A further notable feature of refreezing is its seasonal asymmetry in comparison to melt (Fig. 7), with higher amounts of refreezing occurring earlier in the melt season. This asymmetry causes SNOWPACK to exhibit hysteresis if seasonal runoff is plotted as a function of melt or melt area (not shown). Hysteresis is particular evident for Basin 4, which also indicates the highest asymmetry in seasonal refreezing. Cullather et al. (2016) found a similar hysteresis effect in the regional climate model MAR, which is of comparable complexity as SNOWPACK.

Refreezing rates increase in all basins during 1990–2014 (Fig. 6), but not always at a significant level (Table 1). Generally, the increase in refreezing is restricted to elevations above 1000 m a.s.l. in all regions (Fig. 8). The peak of this increase is around 1500 m a.s.l. for the northern Basins 1, 2 and 8 and at even higher elevations for the more southerly located regions. This increase is primarily caused by a gradual expansion of the melt area to higher elevations, which causes refreezing to occur in formerly dry snow/firn. The increase in refreezing induces both a corresponding decrease in the modelled firn air content and an increase in firn temperature (up to 4° C) due to latent heat release. Particularly for Basins 4 and 5, firn air content decreases also in areas below 1000 m a.s.l.. This reduction is not related to changes in refreezing, but rather caused by increases of liquid surface input and the subsequent transformation of formerly porous firn to bare ice. Increases of firn temperature are





likewise not solely caused by internal heat release due to refreezing but are also affected by changes in skin temperatures at the snow–atmosphere interface. Skin temperatures changes reveal a distinctive spatial variability, with the largest increases in temperature occurring in the northeastern part of the ice sheet, i.e. in the ablation zone of Basin 2. This increase is reflected in the change of the 2–10 m firn temperature in this region (Fig. 8).

The exceptional melt season of 2012 has had an even stronger influence on firn temperatures according to our model simulation: Fig. 9a shows the corresponding refreezing anomaly for this year and Fig. 9b the resulting increase in firn temperature. Almost the entire ice sheet experienced exceptional refreezing rates above the equilibrium line, particularly in the southern area of the GrIS where refreezing anomalies up to 800 kg m$^{-2}$ a$^{-1}$ are modelled. The increase in firn temperature largely reflects the refreezing anomaly, with the strongest warming (6° C and higher) being modelled in the southwestern percolation zone of

the ice sheet. Unfortunately, no observations are available to confirm the pronounced warming. The closest available record is from station KAN_U, which is located in the lower accumulation zone of Basin 6. There, firn temperature increased by approximately 4.7° C during 2012 (Charalampidis et al., 2016). To discuss changes in the vertical firn structure over the simulation period in more detail, we present cross-sections of firn density, temperature and volumetric water content along a southern GrIS transect (Fig. 2) for the beginning of the simulation period (April 1960, Fig. 10) and as relative changes for

the end (April 2014, Fig. 11). In 1960, the transition from bare ice to porous firn is modelled around station KAN_U on the western side of the ice sheet. On the eastern side, no bare ice zone has formed due to the high accumulation rates in this region. Increasing accumulation rates from west to east induce the downward bending of high-density layers east of the ice sheet divide (Fig. 10a). The higher amount of pore space on the eastern side permits larger refreezing fractions, which leads, through release of latent heat, to temperate firn condition close to the margin of the ice sheet and to the formation of a firn aquifer (Fig.

10c).

During the 55 years of the simulation, the firn layer along this transect experienced some distinctive changes: near-surface density increased both on the eastern and western side of the ice sheet (Fig. 11a) with a shift of the transition between bare ice and porous firn to higher elevations on the western side. For 2012, SNOWPACK simulates a bare ice profile for KAN_U while a retrieved density profile for this year revealed layers with porous firn (Charalampidis et al., 2016). Potential causes for this

overestimation in firn density were discussed in Sect. 3.2. Firn temperature increased substantially on both sides of the GrIS but with different patterns (Fig. 11b). In the west, modelled 15 m firn temperature in the percolation zone is relatively stable until 2010 and abruptly increases afterwards by ∼10° C, particularly due to the exceptional melt season of 2012. Refreezing during this year induces a substantial warming of the firn down to a depth of 40 m. At the eastern side, the initial firn temperature is higher by ∼5° C and the simulated warming of the firn is more gradual. A major reason for the less pronounced and shallower

firn temperature increase on the eastern side of the ice sheet is the temporal distribution of liquid input in 2012 (inset panel Fig 11c). On the eastern side, liquid input at the surface is rather evenly distributed throughout the melt season. On the western side, there are several distinctive peaks with liquid input up to ∼40 kg m$^{-2}$ day$^{-1}$. These high fluxes, together with the fact that percolating water is able to bypass cold layers without pore space in our model, cause the relatively deep maximum in firn warming. On the eastern side, the gradual increase in firn temperature allowed the firn aquifer to expand further inland (Fig.

11c), a process that is discussed in more detail in the following section.

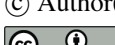



### 4.3  Firn aquifer extent

In the SNOWPACK configuration used here, the model is not able to simulate liquid water ponding on impermeable layers. We therefore apply the term firn aquifer also to temperate firn containing perennial liquid water in unsaturated conditions. Introducing saturated conditions in SNOWPACK would require a definition of the pore space fraction available for liquid

water storage. This quantity is rather uncertain and is assumed to be in the range of 40 % (Jansson et al., 2003) to 100 % (Koenig et al., 2014). For horizontal delineation of modelled firn aquifer area, we use a threshold of 200 kg m$^{-2}$ for the vertically integrated liquid water content in April.

The eastern part of the southern GrIS transect crosses the region where perennial firn aquifers were discovered in 2011 (Forster et al., 2014) and mapped in 2015/2016 (Montgomery et al., 2017). The grey shaded area in Fig. 11 indicates the horizontal

extent of these mapped aquifers. Apparently, the combination of RACMO2.3 and SNOWPACK underestimates the upper elevation of the firn aquifer by approximately 100 m if one assumes only small changes in aquifer extent between 2014 and 2016. A sensitivity test of SNOWPACK with a lower fresh snow density, as described in Sect. 3.2, yields a firn aquifer that reaches higher elevations and thus reduces the mismatch. The reason for this improvement is that the decreased near-surface firn density reduces the conductive heat loss of the aquifer to the atmosphere in winter. The lower boundary of observed firn

aquifers in this area is around 1520 m a.s.l., which coincides with crevasses in the ice stream. It has recently been demonstrated that firn aquifers do not exist in crevassed regions. Apparently, they drain into crevasses, where the water either refreezes or reaches the subglacial drainage system (Poinar et al., 2017). This feature is not included in our model framework, which is why SNOWPACK predicts the presence of aquifers in lower elevated areas. Note that smaller firn aquifers have been mapped downstream of the crevasse fields (Poinar et al., 2017).

Evaluating the modelled depth of the firn aquifer top is difficult, because observations derived from radar measurements return the depth of the water table and not the transition from dry to wet (but unsaturated) firn. Observations returned an average depth of the water table and the aquifer base of 16.2 and 27.7 m, respectively (Montgomery et al., 2017). The depth of the water table is roughly in line with our simulations (Fig. 10 and 11) if one assumes an unsaturated layer between the aquifer top and the dry firn. The depth of the firn aquifer base is overestimated (>40 m) in our simulation, an issue discussed in Steger et al. (2017).

Observations indicate that freezing conditions typically prevail at the base of firn aquifers (Koenig et al., 2014; Montgomery et al., 2017), probably caused by the advection of cold interior ice at greater depths. In SNOWPACK, temperatures below the aquifers are temperate down to the bottom of the model domain, preventing basal refreezing. This could be improved by either initialising the simulations with lower firn temperatures or by applying a non-zero, downward-directed heat flux at the lower boundary of the model domain. However, a more detailed knowledge of the thermodynamic conditions deeper in the firn

would be required to apply such modifications. Due to the present setting of SNOWPACK, which does not allow for saturated condition, the observed mean liquid water content of 16 % (Montgomery et al., 2017) is higher than the modelled values.

Figure 11 shows an expansion of the firn aquifer to higher elevations, which is in line with observations (Montgomery et al., 2017). This expansion is also apparent in other regions, where significant firn aquifer areas are modelled by SNOWPACK (Fig. 12). The highest fractions of firn aquifer area are simulated in the southeastern Basins 4 and 5. In both basins, firn aquifers





considerably expanded inland with time, particularly in Basin 4. This expansion to higher areas is partially compensated by a decrease of aquifer area at lower elevations, where increasing amounts of melt is transforming porous firn to bare ice (Fig. 8). The upward migration of the aquifers in both basins and during the 55 years amounts to ∼200 m in elevation. Upward migration of firn aquifers is also apparent in the other basins, where Basin 3 and 6 reveal a smaller trend of 125 and 90 m, and Basin 8 a higher trend of ∼215 m.

For the entire GrIS, SNOWPACK simulates a firn aquifer area of ∼60,000 km$^2$ for 2014. This value is considerably higher than the estimated value of ∼22,000 km$^2$ by Miège et al. (2016). The discrepancy is likely caused by the different definitions used to delineate firn aquifers. In our study, we use the above-mentioned threshold of 200 kg m$^{-2}$ of liquid water in April to classify model grid cells. In observational derived estimates, firn aquifers are delineated based on the detection of a water table. Thus, our definition also includes areas where firn is not saturated and hence no water table has formed (Steger et al., 2017), as for instance close to the ice sheet margin where aquifers drain into crevasses.

The formation of perennial firn aquifer requires specific climate conditions, i.e. moderate to strong surface melt and high annual accumulation rates (Kuipers Munneke et al., 2014). The dependence of firn aquifers on these parameters is also apparent in our simulation, when modelled GrIS grid cells are plotted as a function of snowfall and liquid input (Fig. 13). The occurrence of firn aquifers is thereby restricted to a rather well separated climatic space, which supports the hypothesis that snowfall and liquid input are the principal predictors for the aquifer formation. The period of 1960–1979 has been selected for this analysis because it is identical to the spin-up period of our simulation. The relation is thus computed for steady-state climate without any long-term trends in the forcing. To assess the influence of a transient climate, firn aquifer occurrence as a function of snowfall and liquid input has also been computed for the period 2005–2014. Apparently, the zone of modelled aquifers shifts to a climatic space with lower snowfall rates. This shift is however likely caused by the changing climate conditions, where the horizontal firn aquifer extent has not yet equilibrated to the new forcing.

## 4.4 Runoff partitioning

Runoff from the ice sheet can either originate from melting of bare ice in the ablation zone, in which case runoff is assumed instantaneous, or from melting of snow/firn in the ablation or accumulation zone, in which case meltwater can be retained or refrozen. Partitioning runoff in these two classes yields insights in basin characteristics and indicates shifts in the accumulation and ablation area extent. Intuitively, basins with high fractions of snow/firn runoff exhibit a higher uncertainty in runoff estimates due to potential storage of liquid water at the location of origin or along the routing path, the latter of which is not explicitly modelled.

Basins 1 and 2 show relatively similar characteristics in terms of runoff partitioning (Fig. 14). In these dry northern regions with relatively wide ablation zones, runoff from snow/firn melt is at least an order of magnitude smaller than runoff from ice melt. Both basins reveal a strong positive trend in runoff from ice between 1990 and 2014 with an increase of ∼30 Gt a$^{-1}$ and ∼16 Gt a$^{-1}$ over the 25 years. The eastern Basin 3 has a higher runoff fraction from snow/firn than the northern basins. Runoff from snow/firn is increasing in the later period, although not on a statistically significant level. A very different picture emerges from Basin 4, which has a very narrow ablation zone. In this basin, approximately 87 % of runoff originates from snow/firn.





Still, there is a significant positive trend in runoff from ice in the second half of the simulation period, which is caused by the decrease of pore space (Fig. 8) and the gradual increase of the ablation zone. As a result, the snow/firn runoff fraction in this basin exhibits a significant negative trend. Basin 5 reveals a similar pattern, but here runoff from ice and snow/firn are comparable in magnitude, particularly towards the end of the simulated period. This basin also shows the highest interannual

variability in snow/firn runoff fraction, which is related to the high interannual variance of winter snowfall in this region. Variance in snowfall is also high in Basin 4, but the sensitivity of the snow/firn runoff fraction on winter snowfall is lower due to a smaller ratio of ablation to accumulation area. The westerly Basins 6–8 exhibit comparable runoff partitionings: all three basins are dominated by runoff from bare ice in the ablation zone and all these fluxes reveal a statistically significant positive trend in the second half of the simulated period. These trends are particularly strong in Basins 6 and 8, where runoff from ice

increases by approximately 55 Gt a$^{-1}$ and 31 Gt a$^{-1}$ over 1990–2014. In the more northerly Basins 7 and 8, there is a small but still significant increase in runoff from snow/firn. For the entire ice sheet, both runoff originating from ice and snow/firn increase at significant rates over the period 1990–2014 by ∼172 Gt a$^{-1}$ and ∼26 Gt a$^{-1}$, respectively. The runoff fraction from snow/firn decreased over this time by approximately 6 %.

## 5 Conclusions

In this study, we analysed a SNOWPACK simulation carried out for the glaciated area of Greenland and for the period 1960–2014 with a focus on the liquid water balance (LWB) of the firn layer. The model was forced by output from the regional atmospheric climate model RACMO2.3 at the upper boundary. A comparison of the modelled cumulative SMB with GRACE and ice discharge data indicates excellent agreement. However, the detrended mean seasonal cycles of these signals reveal significant discrepancies during the melt season. This mismatch can likely be attributed to neglecting runoff transit times and

inaccuracies in the modelled tundra (snow) hydrology. The model also agrees well with observed changes in firn temperature along a 2700 km transect in northwestern Greenland and with firn aquifer occurrence in the southeast. A direct comparison with temperature records from the western percolation zone of the ice sheet is not possible due to an overestimated bare ice zone extent in the model. Among other potential causes, such as climate biases in RACMO2.3, this mismatch is at least partly related to a bias in the fresh snow density parameterisation.

Temporally averaged LWB components over the simulation period (1960–2014) reveal that the balance is dominated by melt, runoff and refreezing in all basins. Modelled changes in retained liquid mass, evaporation and rainfall are typically at least one order of magnitude smaller, even for the more southerly basins. SNOWPACK simulates a mean refreezing fraction of 47 % averaged over the entire ice sheet. This quantity reveals a high spatial variability and is smallest for the northern GrIS (∼30 %) and largest in the southeast (75 %), where snowfall rates are highest. During the first half of the simulation period (1960–

1989), there are no distinctive trends in the components of modelled LWB but this changes for the second half (1990–2014), when surface melt fluxes significantly increase in all basins. In response to this, runoff increases, especially in the southwestern basin of the ice sheet. Simulated trends in runoff generally exceed those in refreezing, which implies that the majority of the additional liquid water input runs off and thus contributes to sea level rise. The only exception is Basin 4 in the southeast,





where most of the additional liquid input is buffered in the firn. The simulated increase in refreezing, which is linked to the gradual expansion of the melt area in all basins, impacts the firn structure by decreasing the firn air content and increasing the firn temperature. The exceptional melt in 2012 in particular causes a substantial warming of the firn, with a peak in the western percolation zone where modelled firn temperatures averaged over 2–10 m depth locally increased by more than 6° C.

SNOWPACK also simulates a migration of the perennial firn aquifer area to higher elevations, which is, at least for an area in southwestern Greenland, in line with observations. Partitioning runoff according to its source (melting ice or snow/firn) shows that runoff from ice dominates on the ice sheet scale ( 78 %), with the highest runoff fractions ($\sim$87 %) from snow/firn modelled in the southeast of the GrIS. Thus, these basins likely exhibit he highest uncertainty in runoff estimates due to possible retention of runoff in snow/firn at the place of origin or along the routing path.

This study also reveals some limitations and inaccuracies of our simulation, which should be addressed in future. First of all, there seems to exist a systematic bias in our applied fresh snow density parameterisation, which yields too high densities for relatively warm conditions. The recently published relation by Langen et al. (2017) could potentially help to improve the bias, but this requires a sound test of the parameterisation in combination with our model for all climate conditions on the GrIS. Secondly, neglecting impermeable layers is problematic for two specific firn conditions: In firn aquifer areas, impermeable ice
layers at the bottom allow ponding of water and thus fully saturated firn, a feature that is not reproduced in our simulation. In the western percolation zone, SNOWPACK simulates near-surface ice layers (Steger et al., 2017), in line with observations. Percolating liquid water is allowed to bypass this potentially cold and non-porous layers in our simulation, where in reality, a certain fraction of the water flux is likely forced to run off laterally above (semi-)impermeable features. How to realistically implement this process, particularly on a horizontal scale of a current RCM, remains however uncertain. Finally, lateral flow of
water is not accounted for in our simulation, which complicates for instance the comparison with GRACE on a seasonal scale. This shortcoming will be addressed in a next step by coupling SNOWPACK to an offline routing scheme for the GrIS.

*Data availability.* All modelled SNOWPACK data presented in this study are available on request from the authors.

*Competing interests.* The authors declare that they have no conflict of interest.

*Acknowledgements.* Christian R. Steger, Carleen H. Reijmer and Michiel R. van den Broeke acknowledge financial support from the Nether-
lands Polar Programme (NPP) of the Netherlands Institute for Scientific Research (NWO) and the Netherlands Earth System Science Centre (NESSC). ECMWF at Reading (UK) is acknowledged for use of the Cray supercomputing system. Graphics were made using Python Matplotlib (version 2.0.0) and Affinity Designer (version 1.5.5).



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




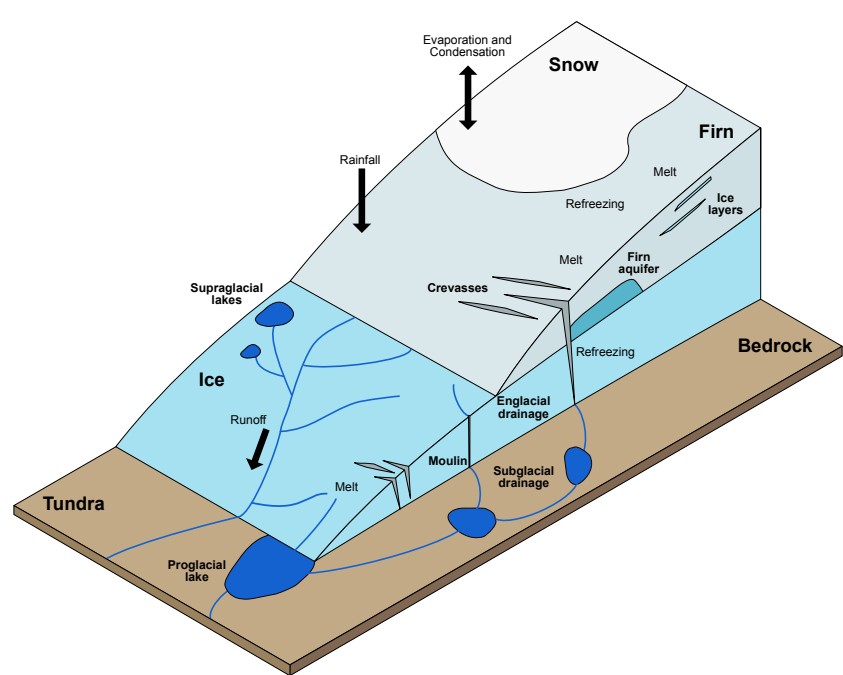

**Figure 1.** GrIS hydrology with the most relevant features and liquid water balance (LWB) components.





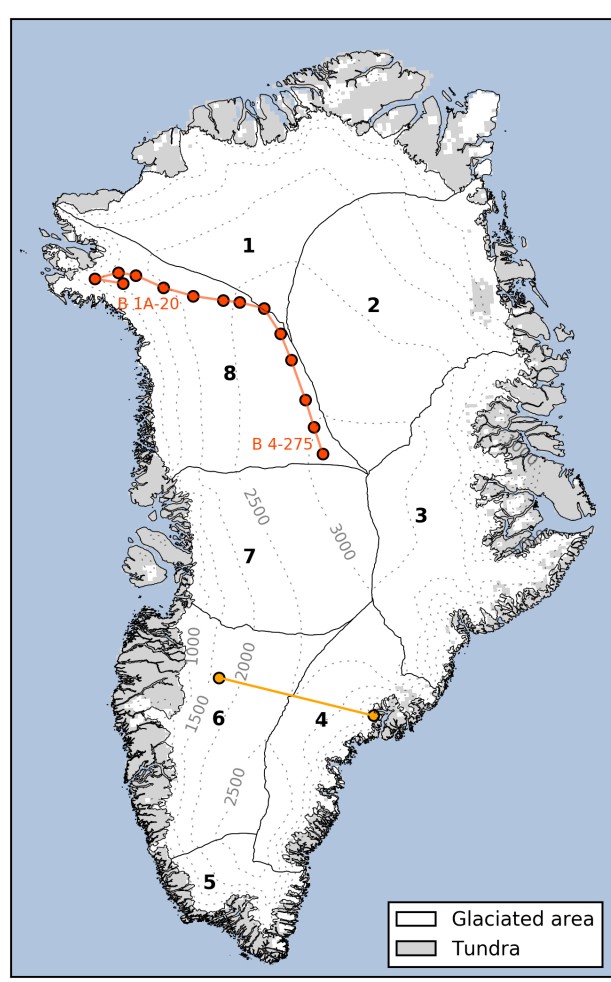

**Figure 2.** Map of Greenland with RACMO2.3 topography (500 m elevation contours as dashed lines) and land surface mask. Thin solid lines delineate eight drainage basins according to Zwally et al. (2012) and the connected circles indicate the locations of firn temperature measurements (red) and the defined southern GrIS transect (orange).





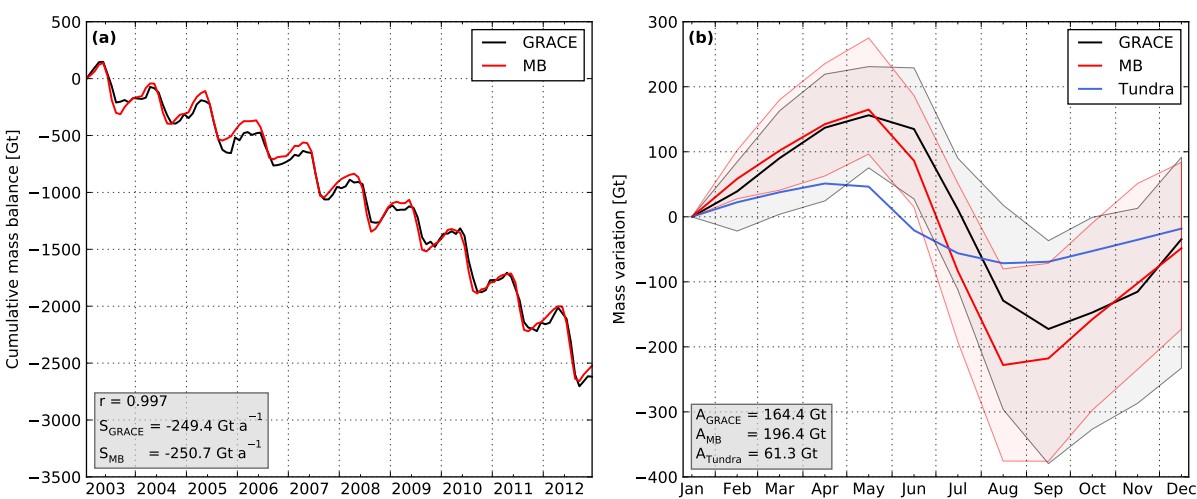

**Figure 3.** Cumulative MB and GRACE time series (a) and detrended seasonal means of these series (b). The inset in (a) shows the correlation coefficient (r) and the slopes (S) of the linear regressions. The inset in (b) provides the mean seasonal amplitudes (A) of the signals and the shaded area illustrates the interannual variability for GRACE and MB.





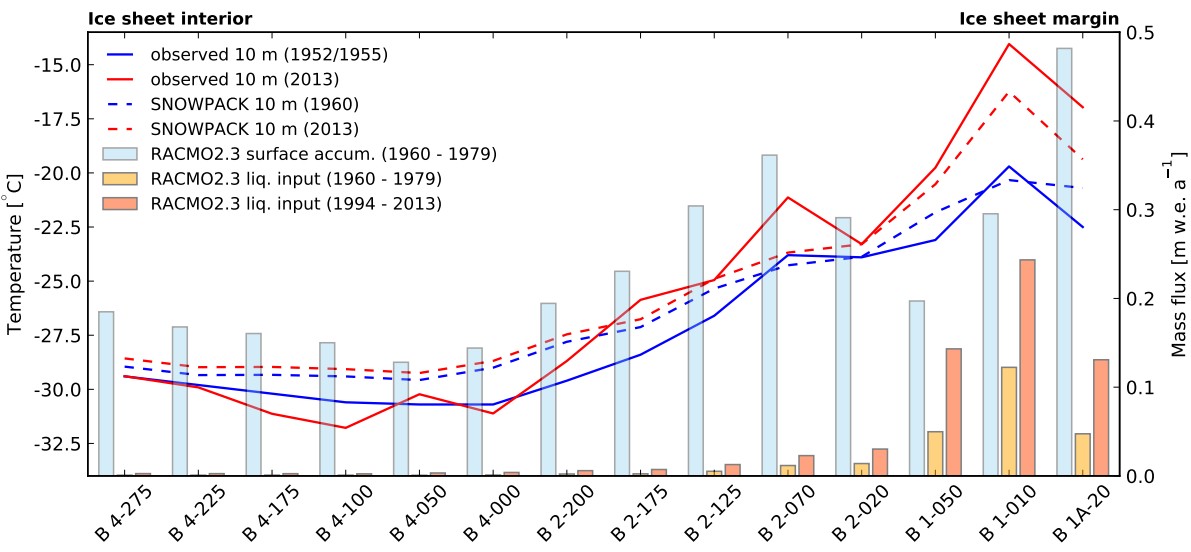

**Figure 4.** Observed and modelled firn temperatures (at 10 m depth) along the NW GrIS transect (Fig. 2). Bars represent RACMO2.3 outputs of surface accumulation and liquid input.





**Figure 5.** Components of the liquid water balance (LWB) for the glaciated area of Greenland averaged over 1960–2014 (a - e). Panel (f) shows refreezing as a fraction of liquid input (rainfall, melt and evaporation). Numbers represent basin-integrated values (excluding peripheral ice caps and glaciers) and the value in the lower right denotes the sum/average for the GrIS. The solid black line marks the mean position of the equilibrium line.





**Figure 6.** Time series of the liquid water balance (LWB) components for the GrIS (top) and the eight basins. Note the different vertical scales. Refreezing fractions in grey represent values between 0 and 100 %.





**Figure 7.** Mean 1960–2014 refreezing as a function of season and elevation. Each cell represents a 7.5 day period and a 100 m elevation bin. Surface melt aggregated with the same method is shown as dashed contour lines and the mean equilibrium line altitude and the elevation of the runoff line are indicated as solid and dashed lines, respectively. The red line displays the elevation bin-averaged firn air content of the upper 40 m.





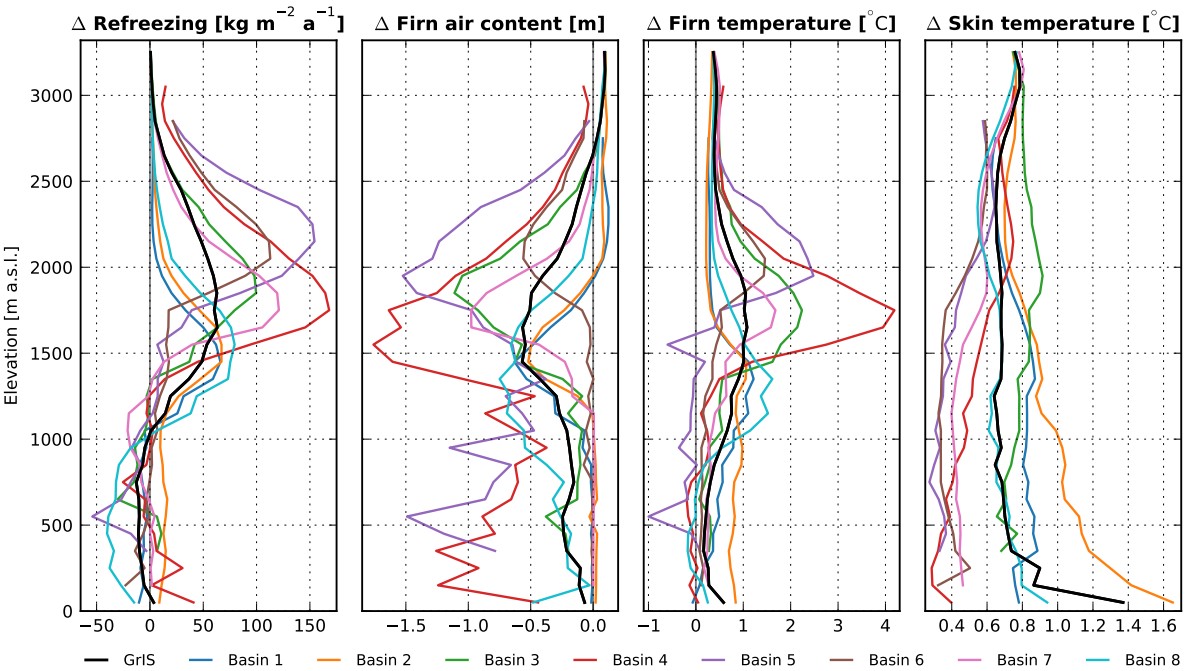

**Figure 8.** Elevation-dependent changes in refreezing, firn air content, firn temperature (averaged over 2–10 m depth) and skin temperature between 1960–1989 and 1990–2014.





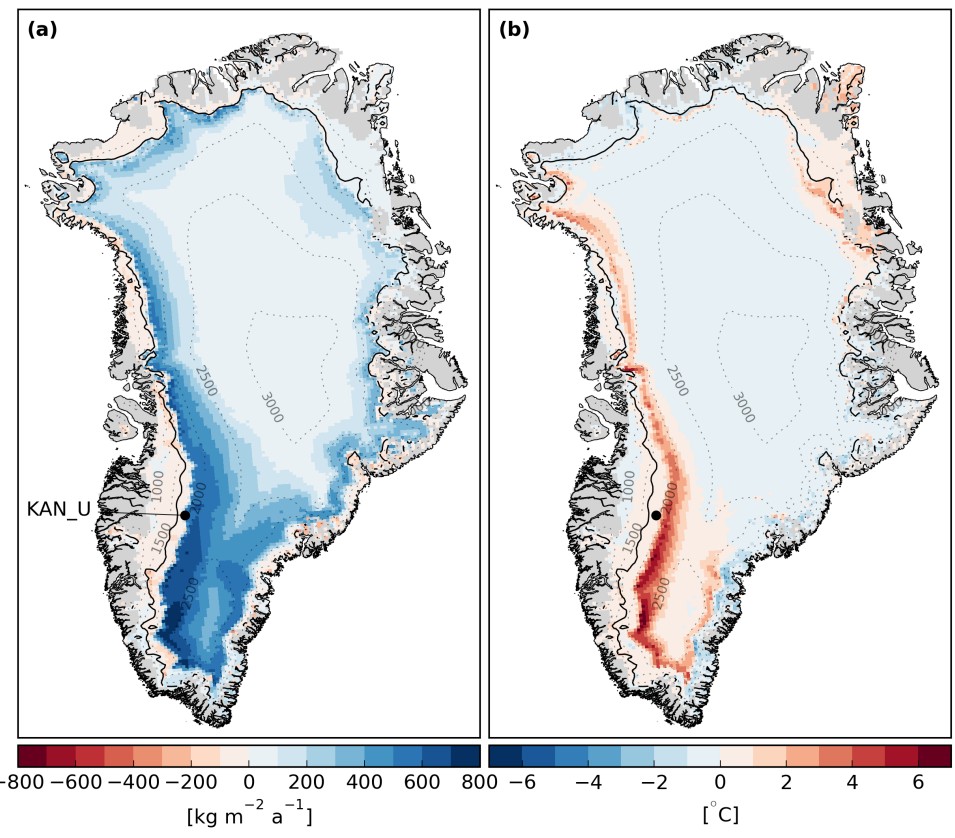

**Figure 9.** Refreezing anomaly of 2012 with reference period 1990–2014 (a) and firn temperature (average over 2–10 m depth) difference between 2011 and 2013 (b). The black line indicates the position of the equilibrium line for the reference period and the black dot the location of station KAN_U.





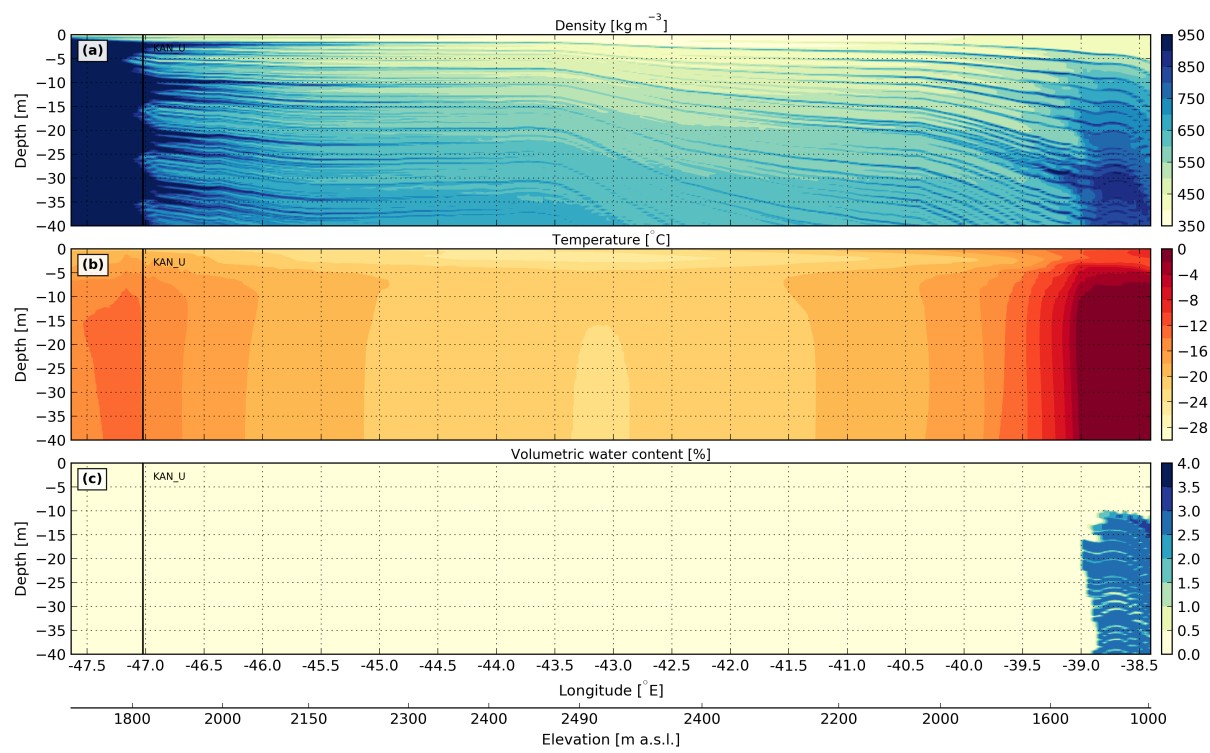

**Figure 10.** Modelled firn properties of the upper 40 m along the southern GrIS transect in April 1960. The vertical black line marks the location of station KAN_U.





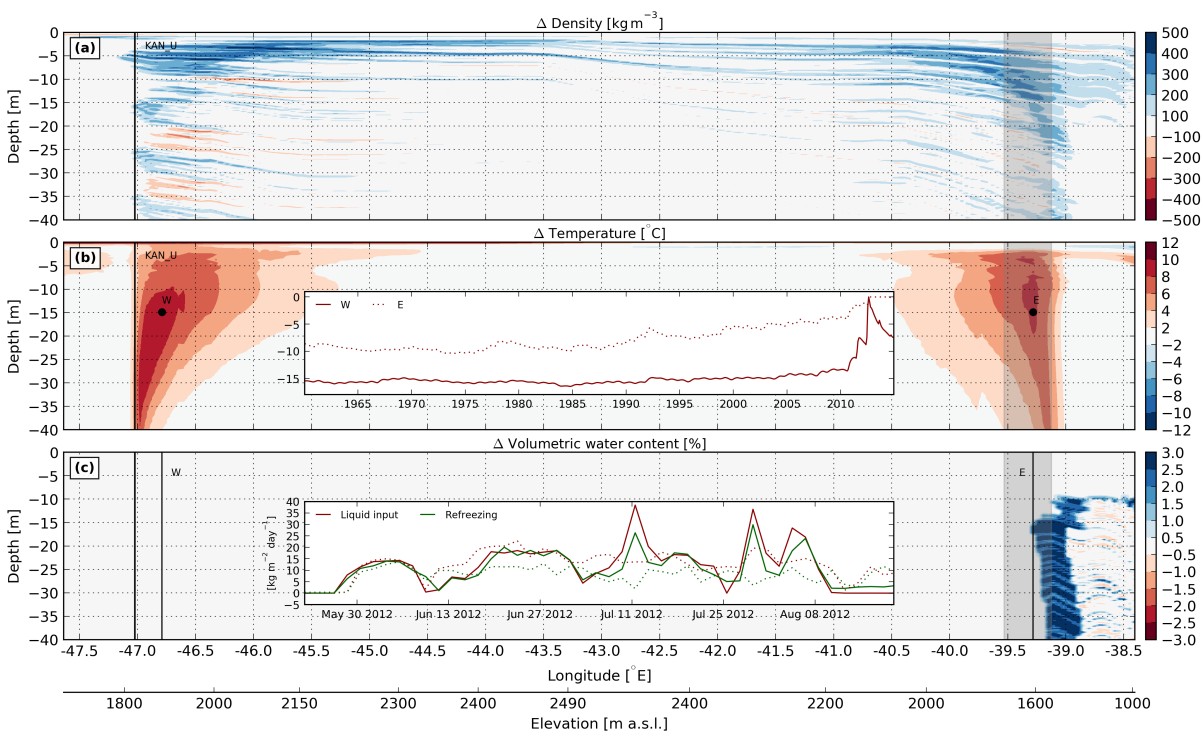

**Figure 11.** Modelled firn property changes of the upper 40 m along the southern GrIS transect between April 1960 and April 2014. The vertical black line marks the location of station KAN_U and the grey shaded area indicates the horizontal extent of observed firn aquifers in 2016 (Montgomery et al., 2017). The inset panel in (b) shows the temporal evolution of firn temperature at the two indicated locations. The inset panel in (c) shows daily mass fluxes of liquid input and refreezing in the summer 2012 for these locations.





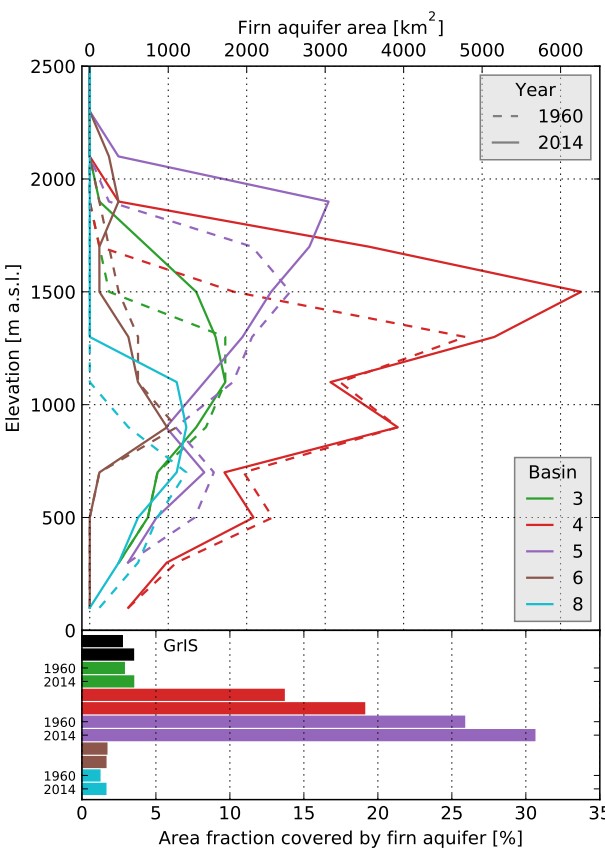

**Figure 12.** Elevation-dependent distribution of firn aquifer areas for the 5 basins where significant aquifers are modelled. Firn aquifer areas are delineated with a liquid water threshold of 200 kg m$^{-2}$ and are aggregated in 200 m elevation bins. In the lower part of the figure, firn aquifer extent is shown as a fraction of the total basin area for the years 1960 and 2014.





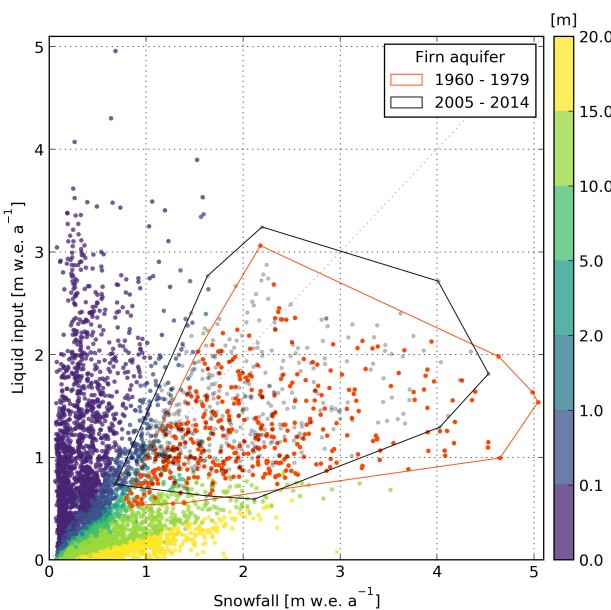

**Figure 13.** Model grid cells with seasonal dry firn as a function of snowfall and liquid input for the period 1960–1979. The colour map shows the firn air content of the upper 40 m for these points. Grid cells with perennial firn aquifer are delineated with a threshold of 200 kg m$^{-2}$ of liquid water and are shown for the period 1960–1979 (orange dots) and 2005–2014 (grey dots).



**Figure 14.** Time series of runoff from ice and snow/firn for the GrIS (top) and the eight basins. The grey shaded area shows runoff from snow/firn as a fraction of total runoff, with values between 0 and 100 %. Dashed lines indicate statistically significant trends (using a significance level of 0.05) between 1990 and 2014.



**Table 1.** Linear trends (1990–2014) in components of the LWB for the GrIS and the eight basins. Statistically insignificant trends (using a significance level of 0.05) are not shown.

| | Melt | Rainfall | Runoff | Refreezing | Refreezing fraction |
|---|---|---|---|---|---|
| | | | $m\,w.e.\,a^{-1}\,(25\,a)^{-1}$ | | $(25\,a)^{-1}$ |
| GrIS | 0.163 | - | 0.115 | 0.052 | -0.086 |
| Basin 1 | 0.145 | - | 0.119 | - | -0.084 |
| Basin 2 | 0.070 | - | 0.051 | - | - |
| Basin 3 | 0.100 | - | 0.057 | - | - |
| Basin 4 | 0.168 | - | 0.050 | 0.117 | - |
| Basin 5 | 0.355 | - | 0.289 | - | - |
| Basin 6 | 0.384 | - | 0.307 | - | - |
| Basin 7 | 0.142 | - | 0.078 | 0.065 | - |
| Basin 8 | 0.169 | 0.014 | 0.136 | 0.046 | -0.158 |