# Peer review of "The modelled liquid water balance of the Greenland Ice Sheet"

_The Cryosphere, 2017_

## Referee Comment (RC1) · Anonymous Referee #1 · 4 Jul 2017

Review Steger et al. (2017). "The modelled liquid water balance of the Greenland ice sheet". The Cryosphere Discussion.

The manuscript addresses the important problem of determining the different contributors to the liquid water balance applied to Greenland. The study highlights the amount of water in the near-surface, using the subsurface scheme from SNOWPACK, which is actually mobilized and available for runoff. SNOWPACK is forced with the regional climate model RACMO2.3, while the manuscript makes a systematically and quantitative comparison of the impact of the different components in the liquid water balance on the Greenland Ice Sheet. The study shows and discusses the significance of changes in some of the critical model parameters to the overall spatial distribution of modelled water retention. However, it fails to sufficiently discuss the implication of the improved

water balance on the surface mass balance.

Overall, this is a decent piece of work, but the manuscript has room for some substantial improvements.

Major points:

1. Highlight differences between Steger et al. (2017) and this study. There seems to be a lot of overlap. Fx fig 4 in Steger et al. (2017) looks almost identical to fig. 5 (e) of this study. It should also be specified if the SNOWPACK model versions and simulations are identical. Also, the firn aquifer description and discussion is very similar.

2. It is stated on p. 6, line 10-11: "At higher elevations in western Greenland SNOW-PACK does simulate a pronounced warming of the firn but there are no in-situ observations available to constrain the magnitude of these changes." The authors should have a look in the extensive GC-net archive of in-situ subsurface temperatures to validate simulated temperatures.

3. Compare simulated refreezing with available firn cores in the literature. However, I believe, that this was done to some extent in Steger et al. (2017)? Please highlight the main outcome of this analysis. How good is the model performing?

4. Define "skin temperature".

5. Quantify statements whenever it is possible throughout the manuscript. For instance statements like "...good model performance...", "...increase in surface melt...", "...indicate positive trends..." or "...temperature increases are highest..." in the Abstract should be quantified. Please have a look at the other sections in the manuscript to quantify similar statements. Please have a look at the Conclusions.

6. The chosen spin up period seems to highly influence crucial subsurface parameters like density (fig 10). This will influence the interpretation of LWB results. Maybe use a different spin up method. Fx the year of 1960 could be used or a mean of the used period (1960-79).

7. Several periods are used for comparing the different components in the liquid water balance. Please highlight/argue why all these periods are used.

8. You should state when the results are presented and discussed.

9. Since you state in the introduction that a better LWB contributes to a better estimate of surface mass balance, how did your experiment modify the surface mass balance? It would be an important point, which is not addressed/discussed sufficiently. Fx Paragraph 4.4 : The description is useful to understand figure 14 but it misses explanation on why those changes in snow/firn/ice melt are relevant for the SMB.

10. In the comparison with grace what is due to the new liquid water balance? How is the liquid water balance influencing the surface mass balance?

11. Basin scale GRACE comparisons with surface mass balance could improve our understanding.

Figures.

Fig. 1: Nice illustrative figure.

Fig. 2: Nice illustrative figure.

Fig. 3: It does not make sense to state a correlation value in Fig 3 unless the time series have undergone a high pass filter, which allow the analysis of the variability of shorter time scales when compared to annual cycle. I.e. the annual cycle should be removed as it will dominate the correlation. Here I would also recommend giving a seasonal R-squared then the mean of that and its standard deviation, which would give a good overview on the mean performance and on its variability.

Fig 4: please add RACMO2.3 surface accum. (1994-2013) to illustrate if changes in accumulation is responsible any temperature changes.

Fig 5: I wonder if the quantities of figs. 5d-f are influenced by the spin up method. Also, the title of (e) and (f) should not be "refreezing" because, I suspect, that the figures

show values of both refrozen and liquid water being retained in the firn. Fx basin 4 is where the perennial aquifers exists seems to be a lot of refreezing there.

Fig 6: Same concern with refreezing vs retention as in Fig. 5.

Fig 7: Same concern with refreezing vs retention as in Figs. 5 and 6.

Fig 8: Please explain in the text, why do you show differences between the periods of 1960-1989 (30 years) and 1990-2014 (25 years). These results could also be influenced be the spin up period.

Fig 9: This is a nice plot, as (a) shows where the firn in 2012 lost its capacity to retain water compared to the reference period. However, I suspect it should be a retention anomaly.

Fig 10: Again, this figure clearly shows the influence of the spin up period. This evident on the western side of the ice sheet with three highly identical subsurface features in the density.

Fig 11: Again, spin up problems?

Fig 12: Nice plot!

Fig 13: Nice plot!

Fig 14: Please explain the implication of changes in the liquid water balance on the modelled runoff pattern in more detail.

Tab 1: I would like to see all trends even if they are not significant.

Specific points:

Line 27 p.3: Which bucket scheme? Please more details and references.

Line 1 on p. 4: It should be mentioned if the two model setups are identical. If not, the differences should be highlighted.

Lines 9-15 on p. 4: More detail is needed for this description of observational data and what is it being used for? Also, many dataset are available (remote sensing, station data, historical and contemporary SMB measurements...) for further validation of the forcing data and of the model output. This part could be improved, which would make the conclusions more solid.

Line 6 on p. 5: How is the tundra hydrology dealt with?

Line 24 on p. 5: It is not only the subsurface temperatures that may be bias but also the density profile.

Line 4 to 7 p. 6: Here and later at KAN-U you mention the overestimation of bare ice zone. A quantification of the spatial extend of this bias would be useful (comparison with remotely sensed bare ice areas?). It would go hand in hand with the many observed SMB available in western Greenland (K transect EGIG line): how does the model compare to them.

Line 6 on p. 6: Who are "they"?

Lines 10-12 on p. 6: Fx, please have a look at GC-Net data.

Line 13 on p. 6: Near-surface snow density depends mostly on wind and subsurface vapor fluxes.

Line 22 on p. 6: Compaction here is mostly due to wind and vapor fluxes.Line 29 on p. 6: Please quantify this inaccuracy?

Line 19 on p. 7: Describe the results from Table 1 in more detail.

Lines 12-16 on p. 8: Should be assessed using observations

Lines 21-25 on p.8: Please explain in more detail the asymmetrical retention pattern and the consequences of this.

Line 10 on p. 9: Again, GC-Net firn temperatures can be used here.

Lines 6-7 on p. 10: How good is this threshold? Where does it come from? Please more support for this is needed.

References:

Steger CR, Reijmer CH, van den Broeke MR, Wever N, Forster RR, Koenig LS, Kuipers Munneke P, Lehning M, Lhermitte S, Ligtenberg SRM, Miège C and Noël BPY (2017). Firn Meltwater Retention on the Greenland Ice Sheet: A Model Comparison. Front. Earth Sci. 5:3. doi: 10.3389/feart.2017.00003

---

## Referee Comment (RC2) · C. M. Stevens (Referee) · 22 Jul 2017

Summary:

This paper presents results of a investigation of the liquid water balance (LWB) of the Greenland Ice Sheet using firn/snow-model (SNOWPACK). The authors' goals are to quantify the components of the LWB spatially and how those quantities have changed in recent decades; to investigate temporal and spatial patterns in refreezing and how those affect the firn; and to assess the models' ability to simulate firn aquifers. The authors force their model using climatic data from the regional climate model RACMO.

Observations of LWB components are unfortunately scarce, but the authors use available data (GRACE data, firn temperatures, firn-aquifer extents) to evaluate their model

performance. The model results compare to GRACE data very well. The model does not reproduce firn temperatures as well, but the results do still compare to the data favorably.

General Comments:

The paper is written and organized well and is a topic of wide interest to the cryospheric-science community. Understanding the LWB is important, as the authors identify, because uncertainty in the runoff component of surface mass balance is a large contributor to sea-level-change estimates. Accurate model simulations are an essential contribution to this scientific issue. Additionally, the authors do a good job of discussing potential sources of model error. I recommend this manuscript for publication with minor revisions.

General points to address:

- The authors mention firn "structure" numerous times. I think to many in the firn community "firn structure" refers to microstructural properties such as grain size, coordination number, etc. In this case, the authors refer specifically to firn temperature and porosity. It may be appropriate to call them by name specifically or use "firn physical properties" as the terminology.

- The authors ignore any lateral flow and also any heterogeneous flow (i.e. piping). I would like a bit more discussion on how those might affect the results, or if it is even possible to know at this point. Section 2.1 asserts that pore space downhill is often filled, but can enough hydraulic head be generated to drive a significant amount of flow? Is there enough data on piping available to do an easy scale analysis of how much heat could be delivered (how deep, how fast?)

- Section 3.2 and figure 4 compares model results to observations. The authors identify biases in the model and discuss lower elevation sites, but what about the higher elevation sites? The model shows a somewhat uniform temperature increase over the

period, but the observations are not as spatially coherent, e.g. sites 4-050 and 4-000. Also, the model gets the lower-elevation structure correct for the modern, but does not as well for 1960. Why is this? Does your assumption that 1952/55 would be the same as 1960 break down (I do think that is a very reasonable assumption, however.)

- Section 3.2 (end): If the different parameterization for fresh-snow density works better, why did you not just use that one? How does this uncertainty affect the results for the higher elevation sites? When you say, "An improved fresh snow density parameterisation seems therefore essential to address this inaccuracy", do you mean an entirely new parameterization is needed, or just a new-to-your-model parameterization? Does the Langen (2017) parameterization fit the criteria of an improved parameterization?

- Section 4.1: Please clarify: You say "changes in the retained liquid mass ($dM_{ret}/dt$) are even smaller...". Equation 1 defines $dM_{ret}/dt$ as the liquid water balance. Is retained liquid mass the same as the LWB? In that sentence, does the quantity in parentheses ($dM_{ret}/dt$) refer to 'changes in retained liquid mass' or to 'retained liquid mass'? I don't doubt the science here but it was confusing to read. If changes in LWB, defined as $dM_{ret}/dt$, are indeed small, then it might imply that is it not an important term in changes in SMB.

- End of page 8, start of page 9,and Figure 8: Please clarify your language: Basin 4 shows a very large increase in firn temperature (due to refreezing I believe), but then you talk about how changes in $t_{skin}$ are also important (but basin 4 does not show increase in $t_{skin}$). Adding a few sentences to clarify this would help – which phenomena is important where?

Specific/technical corrections:

- Use of units throughout: in some places the authors use units of kg/m^2/a (e.g. Figure 5) and in others w.e./a (e.g. Figure 6). It would be good to have consistency throughout. I slightly prefer m w.e./a in this context because it is slightly more intuitive.

- Use of vague language throughout: several instances of "it seems" or "apparently". Just say what you mean directly. E.g. page 5 line 19: change to "it is reasonable".

- Page 1, Line 5: "good model performance" is vague; perhaps "indicate good model-observation agreement" or something along those lines

- P1L7: "increases with" change to "increases at"

- P1L13: be aware that upward could also mean forming at a shallower depth "migration of firn aquifers to higher elevations"

-P1L20-21: put the e.g. section in parentheses to break up the sentence more clearly; change to "and the darkening"

- P1L24: change to "suggest that modeled refreezing"

- P3L15: perhaps a new paragraph at "The Greenland mass . . ."

- P4L1: please clarify: do you mean densification scheme, as in how density changes with time, or the parameterization for new snow density that you use?

- P4L9: get rid of word Futhermore.

- P4L19: it is a bit unclear what you mean by "indirectly". "over" is probably not the best word here.

- P4L25: start new paragraph at "A comparison", rather than at line 28.

- P5L3: do you mean delaying runoff by 18 days in the model? Please clarify.

- P5L7: mean seasonal amplitude in what?

- P5L7: change to "A too-early modeled snow. . ."

- P5L13-14: remove semi-colon and e.g., change to ". . .et al. 2015), or accumulating . . ."

- P5L20-21: change to "Figure 4 shows that SNOWPACK. . ."

- P6L5: I think you have not definded IMAU-FDM acronym prior to this use.

- P6L8: The sentence starting "Due to a different..." is a bit awkward to read – try to rephrase to be active voice.

- P7L12: 47% refreezes – how has this changed since 1960?

- P7L17: list the four most relevant components in parentheses.

- P7L20: "Remarkably" – why is this remarkable? Would you have expected other regions to also have more rain? Is there more precipitation in total in that region, or higher rainfall as a percentage of total precipitation – how does that compare to other areas?

- P7L27: change "does" to "do"

- P7L31: change "one" to "trend"

- P8L6: Is melt climate? I might suggest that temperature is climate, and melt is a result (but maybe I am wrong or nit-picking or both)

- P8L13: do you mean "the seasonal decrease in refreezing at higher elevations is caused by ..."?

- P8L23-26: Are these sentences about hysteresis needed? They seem distracting and not relevant to me, but if they are relevant, include some discussion about how it affects your results and conclusions.

- P8L29: get rid of word "even"

- P8L30: change to "causes melt and refreezing..."

- P9L18: change "higher" to "larger"

- P9L18 and Figure 10: perhaps adding a panel to figure 10 to show pore space across the transect?

- P10L10: get rid of word "apparently"

- P10L16: get rid of apparently, say something like, "Instead, in these regions liquid water drains into . . ."

- P10L20: Change to: "Comparing the modeled depth of the firn-aquifer top to observations is difficult . . ."

- P10L23: change to "unsaturated wet layer"

- P10L26: unclear what you mean by advection of cold interior ice here – please expand on the physics of what is going on.

- P10L28: would initializing the model with lower temperatures be a physical thing do to? Is there any reason to believe that firn temperatures were lower in 1960 (assuming that is the start of the simulation)?

- P10L32: Specify the time period over which that expansion is occurring.

- P11L4-5: are these trends or just changes?

- P11L11: hyphen in ice-sheet margin

- P11L12: s in aquifers; change "i.e." to "specifically"

- P11L14: no comma after simulation

- P11L16: get rid of word "the" before aquifer formation

- P11L26: get rid of word "intuitively"

- P13, end: also mention piping perhaps

- Figure 3 caption: change to "shaded areas illustrate"

- Figure 5: It might be nice to have basin numbers labeled or on this figure to avoid flipping to figure 2. Why is refreezing the only parameter you show fraction for? You could, for each LWB component, show the value (as you do) with the fraction of total in

parentheses. So, for example, basin 1 in panel e would have the value 14 (30%). That way the reader could see the fraction for all of the components. Also on this figure: The outlines of the basins are tough to see (make them darker and make the ELA a dashed dark line?). The color scales make it tough to see what is going on near the margins – not sure how to fix that.

- Figure 6: the legend could be larger at the bottom, especially with the thickness of the lines

- Figure 8: Perhaps move the x-labels to the bottom so that the numbers and labels are next to one another. It is a bit confusing – is this showing the change in the average values for 1960-1989 subtracted from the average values for 1990 – 2014? Also on this figure: legend could be larger

- Figure 10: An additional panel showing the elevation/surface profile of the ice sheet could be useful here.

- Figure 11: inset panels are quite small.

- Figure 13: Put a label along the color bar: Firn Air Content (m). The dots here make this a bit tough to understand. Perhaps making the aquifer dots more contrasting colors? Or use crosses? The grey dots are tough to see.

- Table 1: The units are difficult to understand here. Does m w.e. aˆ-1 (25a)ˆ-1 refer to melt, rainfall, runoff, and refreezing? Since you state in the caption that these trends are 1990 – 2014, does it not suffice to say the trend is in m w.e. per year?

---

## Author Comment (AC1) · 22 Aug 2017

Dear Anonymous Referee #1, please find attached our response and a revised manuscript. Sincerely, Christian Steger

Please also note the supplement to this comment:
https://www.the-cryosphere-discuss.net/tc-2017-88/tc-2017-88-AC1-supplement.pdf

---

## Author Comment (AC2) · 22 Aug 2017

Dear reviewers,

Thank you very much for the constructive and detailed comments and suggestions to the manuscript. Please find below our responses to the individual points. To facilitate readability, responses are in blue and modifications in the manuscript in red.

**Anonymous Referee #1**

The manuscript addresses the important problem of determining the different contributors to the liquid water balance applied to Greenland. The study highlights the amount of water in the near-surface, using the subsurface scheme from SNOWPACK, which is actually mobilized and available for runoff. SNOWPACK is forced with the regional climate model RACMO2.3, while the manuscript makes a systematically and quantitative comparison of the impact of the different components in the liquid water balance on the Greenland Ice Sheet. The study shows and discusses the significance of changes in some of the critical model parameters to the overall spatial distribution of modelled water retention. However, it fails to sufficiently discuss the implication of the improved water balance on the surface mass balance.

Overall, this is a decent piece of work, but the manuscript has room for some substantial improvements.

**Major points:**

1. Highlight differences between Steger et al. (2017) and this study. There seems to be a lot of overlap. Fx fig 4 in Steger et al. (2017) looks almost identical to fig. 5 (e) of this study. It should also be specified if the SNOWPACK model versions and simulations are identical. Also, the firn aquifer description and discussion is very similar.

Figure 5 (e) is indeed almost identical to Fig. 4 in Steger et al. (2017) but we think it is useful to present it again to give a complete picture of the spatial patterns of the most relevant LWB components.
The model version of SNOWPACK was added to the manuscript (Sect. 2.2) and the end of this section was extended to state that the SNOWPACK runs are identical:
A more detailed description of the model setup and the applied spin-up procedure is stated in Steger et al. (2017), where the same SNOWPACK run was used.
Throughout the manuscript we removed redundant statements, particularly in the firn aquifer section and in the conclusions. However, we kept e.g. the discussion about the vertical extent of firn aquifers in view of new available observational data (Montgomery et al., 2017).

2. It is stated on p. 6, line 10-11: "At higher elevations in western Greenland SNOWPACK does simulate a pronounced warming of the firn but there are no in-situ observations available to constrain the magnitude of these changes." The authors should have a look in the extensive GC-net archive of in-situ subsurface temperatures to validate simulated temperatures.

We thank the reviewer for pointing out this data archive. We looked into subsurface temperature data from relevant stations (DYE-2, Crawford Point 1 & 2 and GITS) but the data of (at least) the recent decade is either missing or of very poor quality (large data gaps and/or unphysical high-frequency fluctuations). This is likely caused by the steady progression of sensor deterioration (personal communication with Konrad Steffen). It is therefore not possible to validate the recent simulated firn warming in the lower accumulation zone of the western GrIS with this data set. We added the following sentences to the manuscript to make this clear:
Other snow/firn temperature records are available from the Greenland Climate Network (GC–Net; Steffen and Box 2001) and for the western percolation zone (Humphrey et al., 2012; Charalampidis et al., 2016). [...] Unfortunately, the subsurface temperature data recorded at GC-Net stations located in this area (DYE-2, Crawford Point 1 & 2 and GITS) suffer from large data gaps and/or unphysical high-frequency fluctuations caused by sensor deterioration (K. Steffen, personal communication). The data are thus of insufficient quality to verify these changes.

3. Compare simulated refreezing with available firn cores in the literature. However, I believe, that this was done to some extent in Steger et al. (2017)? Please highlight the main outcome of this analysis. How good is the model performing?

We do not fully understand the reviewer's intention of comparing simulated refreezing with available firn cores. A quantity that is compared to observations in Steger et al. (2017) is snow/firn density, which is a combined result of compaction and refreezing. In terms of firn density, SNOWPACK indicates generally a better performance than the IMAU-FDM - particularly for comparably warmer climatic conditions.
We added to the manuscript:
Although modelled refreezing cannot directly be evaluated with observations, Steger et al. (2017) made a comprehensive assessment of modelled firn density, which is the combined result of dry compaction and refreezing. Results show a reasonable performance of SNOWPACK, but a general overestimation of densities in the percolation zone. This bias is likely the result of overestimated near-surface refreezing caused by neglecting heterogeneous water percolation, an overestimation of fresh snow density and errors in the atmospheric forcing (Steger et al., 2017).

4. Define "skin temperature".

Skin temperature is the equilibrium temperature of an infinitesimally thin layer without heat capacity, which represents the interface between the atmosphere and the ground. It is often used interchangeably with surface temperature. We stated this in the manuscript:
Skin temperature is the temperature of an infinitesimally thin layer without heat capacity, and is representative for surface temperature.
And we now use surface temperature instead of skin temperature throughout the script, because it is a more commonly used term.

5. Quantify statements whenever it is possible throughout the manuscript. For instance, statements like "...good model performance...", "...increase in surface melt...", ". . .indicate positive trends. . ." or ". . .temperature increases are highest. . ." in the Abstract should be quantified. Please have a look at the other sections in the manuscript to quantify similar statements. Please have a look at the Conclusions.

We checked all sections of the manuscript and supported statements with numerical values where applicable/possible. Some statements are rather general (particularly in the abstract) and are thus difficult to support with numerical quantities, because providing such number would require a detailed explanation about their validity (e.g. firn temperature increase → where exactly, over which temporal period, averaged over what depth, etc.)

6. The chosen spin up period seems to highly influence crucial subsurface parameters like density (fig 10). This will influence the interpretation of LWB results. Maybe use a different spin up method. Fx the year of 1960 could be used or a mean of the used period (1960-79).

Indeed, the spin-up is an important factor in our simulation that influences the modelled subsurface properties. In the absence of firn profile measurements or knowledge about the climatological conditions prior to the start of the simulation, one has to generate the firn layer with meteorological data from the simulation period (and assume that the prior climate was comparable to the one at the beginning of the simulation period). The definition (length, average, etc.) of the spin-up period is debatable:
As the reviewer suggests, one way is to repeatedly loop over the first year (in our case 1960), as done in other studies (Mottram et al., 2017; Schmidt et. al., 2017). However, an issue of this approach is the assumption that the chosen year is representative for the climate prior to the simulation start. By doing this, one neglects the fact that this year may substantially deviate from the mean climate and the interannual variability of the climate is also not captured. Hence, we think that it is more appropriate to use a longer time span for the spin-up. The selected time span of 20 years is long enough to capture interannual variability and it is located in a period where the GrIS experienced a relatively stable climate. As the reviewer states, one could also compute an average over this 20 years for the spin-up. However, we are not convinced that this would improve the spin-up because on the one hand, one would lose interannual variability (which is present in "real" firn profiles), and on the other hand, one may create inconsistencies in the forcing data.

We are aware of the shortcomings of our approach (e.g. the cyclic occurrence of high density layers from years with intensive surface melt), but considering the above-mentioned facts, we do not see a distinctively better technique to perform the spin-up.

7. Several periods are used for comparing the different components in the liquid water balance. Please highlight/argue why all these periods are used.

In the model evaluation part, the chosen periods are constrained by the temporal availability of observational data. In the comparison of modelled and measured firn temperatures, we provide RACMO2.3 surface mass fluxes, which are averaged over the 20 years prior to the temperature measurements. Due to the lack of model data before 1960, we assume a constant climate and provide averaged mass fluxes over 1960–1979 for the temperature records of 1952/1955.
For the majority of the LWB climatology, we use the two periods (1960–1989) and (1990–2014). This partitioning is very similar to the one applied in Van den Broeke et al. (2016) and it divides the entire simulated period in a first part with a rather constant climate and a second part with a distinct increase in melt. Additionally, we show averages over the entire period (1960–2014) when we discuss mean characteristics.
In Fig. 9, we illustrate the 2012 anomaly because this year was exceptional in terms of melt extent and amount. In the following discussion of changes in vertical firn properties and firn aquifers, we illustrate changes over the entire simulation period and thus select the years 1960 and 2014.
In Fig. 13, the first period is identical to the spin-up period of the model to illustrate the occurrence of firn aquifers in a steady-state climate. The second period was somehow arbitrary selected and is now changed to (2010–2014), which is identical to the period of firn aquifers observations by remote sensing (Miège et al., 2016).
We added the temporal availability of the ice discharge and GRACE data to Sect. 2.3 to elaborate on the selected temporal period for computing the MB (2003–2012). The reason for using the two periods (1960–1989 and 1990–2014) is briefly explained in Sect. 4.1. In Sect. 4.2, we state why we selected the year 2012 for a more detailed analysis in terms of refreezing and firn temperature change. Additionally, we changed the following sentence to explain the second period (2010–2014) used for the evaluation of firn aquifer occurrence:
To assess the influence of a transient climate, firn aquifer occurrence as a function of snowfall and liquid input has also been computed for the period 2010–2014, which is identical to firn aquifer observations by remote sensing (Miège et al., 2016).

8. You should state when the results are presented and discussed.

We now state at the end of Sect. 1 (Introduction) the beginning of the results and discussion sections by changing the following sentence:
Subsequently, we discuss the comparison of model output with remote sensing data (GRACE) and in situ measurements (firn temperatures). Section 4 contains the results of the LWB evaluation and a more detailed analysis of refreezing, runoff and changes in different firn properties.

9. Since you state in the introduction that a better LWB contributes to a better estimate of surface mass balance, how did your experiment modify the surface mass balance? It would be an important point, which is not addressed/discussed sufficiently. Fx Paragraph 4.4: The description is useful to understand figure 14 but it misses explanation on why those changes in snow/firn/ice melt are relevant for the SMB.

Please see answer to question 10 (major points).

10. In the comparison with grace what is due to the new liquid water balance? How is the liquid water balance influencing the surface mass balance?

In order to highlight the link between the LWB and the SMB, we also included the definition of the SMB in Sect. 2.1 and briefly mention the common components.
Considering the LWB, refreezing and runoff are the quantities that are most different between SNOWPACK and RACMO2.3's own snow model. In regions with high melt, SNOWPACK tends to simulate lower and more realistic snow/firn densities (Steger et al., 2017). This allows, together with the higher value of prescribed irreducible water, for more refreezing (and thus less runoff). To illustrate how this affects SMB, MB computed entirely with RACMO2.3 data is now included in Fig. 3.

The higher refreezing and lower runoff rates in SNOWPACK lead to a more positive SMB and thus to a reduced decline in cumulative MB. This improves the agreement with GRACE. We added the following text to the manuscript (Sect. 3.1) to discuss this:

A comparison between the derived cumulative MB and GRACE is provided in Fig. 3a. The MB is computed by taking the simulated SMB over the glaciated area either from RACMO2.3 or SNOWPACK. Both cumulative MBs indicate an excellent agreement with GRACE ($R^2 > 0.99$). In terms of linear trends, SNOWPACK agrees better with GRACE due to higher modelled refreezing fractions and thus lower amounts of runoff from the ice sheet. [...] The minima in the MBs occur both earlier and with higher magnitudes than in GRACE, where SNOWPACK performs slightly better due to smaller amounts of modelled runoff.

11. Basin scale GRACE comparisons with surface mass balance could improve our understanding.

As the reviewer states correctly, a basin-scale comparison would enhance our understanding of the modelled SMB. However, despite the availability of GRACE products on basin scales, we omit such a comparison because we suspect significant signal leakage in GRACE on the spatial scale of the eight basins. Hence, we are not convinced that a comparison of the trends (of GRACE and SMB – ice discharge) on a basin scale would yield meaningful additional insights.

**Figures:**

Fig. 1: Nice illustrative figure.

Thanks.

Fig. 2: Nice illustrative figure.

Thanks again.

Fig. 3: It does not make sense to state a correlation value in Fig 3 unless the time series have undergone a high pass filter, which allow the analysis of the variability of shorter time scales when compared to annual cycle. I.e. the annual cycle should be removed, as it will dominate the correlation. Here I would also recommend giving a seasonal R-squared then the mean of that and its standard deviation, which would give a good overview on the mean performance and on its variability.

We do agree with the reviewer's opinion that applying a high pass filter to the time series would be necessary to assess variability on shorter time scales than the annual cycle. However, the trend as well as the annual cycle are the signal components we actually want to compare between GRACE and the MB. Therefore, we do think it is meaningful to compute a direct correlation between the time series.
We did not incorporate the reviewer's suggestion of computing a seasonal R-squared as we were not able to understand the intended method entirely.

Fig 4: please add RACMO2.3 surface accum. (1994-2013) to illustrate if changes in accumulation is responsible any temperature changes.

We added RACMO2.3 surface accumulation (1994–2013) to Fig. 4 and added the following sentence to Sect. 3.2:

Along the entire transect, modelled increases in solid precipitation are spatially rather uniform and small (~0.02 m w.e. a$^{-1}$), and therefore likely less relevant for explaining changes in firn temperature.

Fig 5: I wonder if the quantities of figs. 5d-f are influenced by the spin up method. Also, the title of (e) and (f) should not be "refreezing" because, I suspect, that the figures show values of both refrozen and liquid water being retained in the firn. Fx basin 4 is where the perennial aquifers exist seems to be a lot of refreezing there.

The dependency of runoff on the spin-up method chosen is rather small for the majority of the GrIS, as runoff mostly originates from bare ice (Fig. 14). Part of the runoff from snow/firn originates from seasonal snow over bare ice in the ablation zone and is therefore not influenced by the spin-up either.

In Basins 4 and 5 however, a considerable amount of runoff originates from porous snow/firn. These areas are more sensitive to the selected spin-up method but other ill-constrained factors, such as vertical firn temperature initialisation and heat flux description at the bottom of the model domain, will outweigh these uncertainties because the accumulation rate rapidly refreshes the firn layer. Refreezing likely exhibits the strongest dependency on the chosen spin-up method, but because of the reason mentioned above firn porosity and cold content mainly depend on the recent climate and thus we do not expect substantially modified refreezing rates if the spin-up method were changed. The titles of (e) and (f) are correct and only refreezing rates are displayed, i.e. not the combined effect of refreezing and liquid water retention.
To clarify this, we added the following sentences to Sect. 2.2:
Neglecting heterogeneous percolation causes refreezing to occur mostly in the upper snowpack, where temperature and porosity are determined by the recent climate.

Fig 6: Same concern with refreezing vs. retention as in Fig. 5.

Here, also, only refreezing rates are displayed (not combined with liquid water retention in the firn).

Fig 7: Same concern with refreezing vs. retention as in Figs. 5 and 6.

Please see previous answers.

Fig 8: Please explain in the text, why do you show differences between the periods of 1960-1989 (30 years) and 1990-2014 (25 years). These results could also be influenced be the spin up period.

See answer to question 6 and 7 (major points): the main reason is to have a reference period with a relatively constant climate and a period with rapid change, both having comparable lengths.

Fig 9: This is a nice plot, as (a) shows where the firn in 2012 lost its capacity to retain water compared to the reference period. However, I suspect it should be a retention anomaly.

The plot shows only refrozen mass (not combined with liquid water retained in the firn).

Fig 10: Again, this figure clearly shows the influence of the spin up period. This is evident on the western side of the ice sheet with three highly identical subsurface features in the density.

True, please see answer to question 6 (major points).

Fig 11: Again, spin up problems?

Please see answer to question 6 (major points). It is also interesting to note that this only becomes evident close to the equilibrium line where accumulation is small.

Fig 12: Nice plot!

Thanks.

Fig 13: Nice plot!

Thanks.

Fig 14: Please explain the implication of changes in the liquid water balance on the modelled runoff pattern in more detail.

Please see answer to question 10 (major points).

Tab 1: I would like to see all trends even if they are not significant.

We now show all trends and marked the insignificant ones by an asterisk.

**Specific points:**

Line 27 p.3: Which bucket scheme? Please more details and references.

We added two references to the manuscript (in Sect. 2.2), in which the bucket scheme used in SNOWPACK is described.

Line 1 on p. 4: It should be mentioned if the two model setups are identical. If not, the differences should be highlighted.

Please see answer to question 1 (major points).

Lines 9-15 on p. 4: More detail is needed for this description of observational data and what is it being used for? Also, many datasets are available (remote sensing, station data, historical and contemporary SMB measurements...) for further validation of the forcing data and of the model output. This part could be improved, which would make the conclusions more solid.

The RACMO2.3 output, used as forcing data in this study, was already extensively evaluated in the study by Noël et al. (2015). We have now made this more explicit in the manuscript by adding the following sentence to Sect. 2.2:
The capability of RACMO2.3 to accurately simulate present-day surface climate on the GrIS was illustrated in an extensive evaluation by Noël et al. (2015).
SNOWPACK's performance in terms of snow/firn density and firn aquifer extent was assessed in the Steger et al. (2017). In this manuscript, we additionally evaluated the spatially integrated SMB of SNOWPACK (forced with RACMO2.3) and we compared observed and modelled firn temperatures (where measurements are available and a comparison is meaningful). We agree with the reviewer that further evaluations of SNOWPACK would be interesting, particularly for subsurface processes as refreezing, but we think that we exploited all presently available/suitable data sources.

Line 6 on p. 5: How is the tundra hydrology dealt with?

We kindly refer the reviewer to the first paragraph of Sect. 3.1 where the RACMO2.3 tundra snow model is described, and to the last paragraph of this section where we list relevant features of the tundra hydrology. These are currently not considered in our model framework.

Line 24 on p. 5: It is not only the subsurface temperatures that may be bias but also the density profile.

Good point; we included this consideration by changing the sentence to:
The bias for the second period is more difficult to explain in the absence of continuous firn temperature measurements and firn density records.

Line 4 to 7 p. 6: Here and later at KAN-U you mention the overestimation of bare ice zone. A quantification of the spatial extend of this bias would be useful (comparison with remotely sensed bare ice areas?). It would go hand in hand with the many observed SMB available in western Greenland (K transect EGIG line): how does the model compare to them.

A qualitative picture of this overestimation can be found in Steger et al. (2017) (Fig. 9), where vertical firn density transects of the IMAU-FDM and SNOWPACK are compared with data from the NASA Operation IceBridge accumulation radar. The figure confirms the overestimation of the bare ice zone by both models but it does not allow for an exact quantification of the bias.
A quantification of the bare ice zone with remote sensing data (by distinguishing between different surface properties of snow/firn and ice), as probably intended by the reviewer, could be performed. However, delineating ice and snow/firn areas with such a technique could result in a substantial overestimation of the bare ice zone due to a misclassification of porous firn covered by near-surface ice (Machguth et al., 2016). We consider such an effort beyond the scope of the present paper. Therefore, we restrict our evaluation to a comparison of modelled SMB values along the K-transect:

[Figure]

The above figure shows SMB components of the different models compared to stake measurements along the K-transect (average 1990–2014). All models return similar SMB values, which indicates that the forcing data by RACMO2.3 dominates the modelled SMB values rather than the individual snow/firn models, at least for this transect.

The values of RACMO2.3 and SNOWPACK are almost identical and there is only one larger difference around S8, which is caused by the interpolation of SNOWPACK data from the checkerboard to the full RACMO2.3 grid (Steger et al., 2017). The IMAU-FDM has higher runoff and therefore lower SMB values around S10 compared to the other models, for reasons explained earlier. Generally, modelled SMB values are in close agreement with stake measurements between S9 and S10, where the transition of bare ice to porous firn is located. However, there is a fundamental problem of comparing stake measurements to modelled SMB values at locations with porous firn: Stake measurements only capture mass changes at the surface. The modelled SMB as presented in this paper however considers vertically integrated mass changes (→ climatic SMB). I. e. the climatic SMB at S10 could be higher than the one derived by stake measurements due to refreezing of mass deeper in the firn.

We added the following sentence to the manuscript to state the difficulty of using remote sensing data to quantify the bare ice zone extent:

Inferring the bare ice zone from remote sensing data, e.g. by using the different surface properties of snow and ice, is complicated due to formation of near-surface ice layer (Machguth et al., 2016) above porous firn.

Line 6 on p. 6: Who are "they"?

"They" refers to the models. The sentence was reformulated to remove this ambiguity:
The reason is the overestimation of the bare ice zone on the western GrIS by the IMAU-FDM and SNOWPACK; i.e. the models are incapable of simulating the subsurface warming due to a deficiency of pore space for refreezing.

Lines 10-12 on p. 6: Fx, please have a look at GC-Net data.

Please see answer to question 2 (major points).

Line 13 on p. 6: Near-surface snow density depends mostly on wind and subsurface vapour fluxes.

We agree with the reviewer that wind and subsurface vapour fluxes are relevant factors for the near-surface density. Instead of adapting our sentence, we decided to remove it because it didn't really connect this paragraph to the previous one. We added a new sentence to accomplish that:

To address the above-mentioned model bias in overestimating the bare ice zone, we briefly assessed fresh snow density, which is a rather uncertain factor in our simulation.

Line 22 on p. 6: Compaction here is mostly due to wind and vapour fluxes. Line 29 on p. 6: Please quantify this inaccuracy?

We agree with the reviewer that compaction due to wind and vapour fluxes are the most important factors for near-surface snow densification. However, these effects are already taken into account in the fresh snow density parameterisation we apply. To clarify, we wrote:
The parameterisation, which accounts for near-surface densification due to wind and vapour fluxes, clearly overestimates fresh snow density for this region.
And we removed the part about overburden-dependent compaction, because it is indeed of minor relevance in the uppermost 50 cm compared to other process that induce densification.

Line 19 on p. 7: Describe the results from Table 1 in more detail.

We extended the description about linear trends (1990–2014) in the manuscript (Sect. 4.1) - particularly for changes in melt and rainfall:
Changes are particularly large for Basin 5 and 6, where melt increases by 0.36 m w.e. a$^{-1}$ and 0.38 m w.e. a$^{-1}$, respectively. The dominant cause for these large changes is the comparably high increase of melt in the ablation area of the GrIS, especially in the southwest. Modelled snow melt in the ablation zone is particularly sensitive to temperature increases due to the albedo difference between snow and ice, where bare ice with a lower albedo is more rapidly exposed through accelerated melt of snow. The lowered surface albedo subsequently enhances melt of bare ice. A secondary cause is the relatively flat hypsometry of these basins, where 58 % respectively 47 % of the area is below 2000 m a.s.l. (compared to 39 % for the GrIS). Rainfall, as a further contributor to liquid input, does not exhibit a significant trend for the majority of the basins. Linear trends are comparably high for Basin 5 (1.22 mm w.e. a$^{-2}$) and Basin 6 (0.43 mm w.e. a$^{-2}$) but statistically insignificant. Remarkably, the northwestern Basin 8 is the only region with a significant positive trend in rainfall of 0.56 mm w.e. a$^{-2}$. This increase is not caused by a change in total precipitation but by a significant increase of the rainfall fraction in this area.

Lines 12-16 on p. 8: Should be assessed using observations

The figure below shows the temporal evolution of firn temperature between April 2013 and March 2014 for a firn aquifer location in southeast Greenland (we thank Clément Miège for proving this data).

[Figure]

Grey areas mark missing data and it should be noted that the vertical coordinate always refers to the initial depth of the temperature sensors ($\rightarrow$ no correction for surface accumulation/ablation). It is evident that the vertically integrated cold content of the firn layer steadily decreases from April to June until the entire firn column is isothermal at 0° C in early August. Near-surface temperatures (~ upmost 0.5 m) are at the melting point between mid-June and mid-August due to high incoming shortwave

radiation and/or downward directed sensible heat fluxes. Additionally, it is likely that the release of latent heat from refreezing (particularly during the night) keeps the near-surface temperature constantly at 0° C. However, to verify this, one would need temperature observations with a high temporal resolution (e.g. hourly) and with a sensor that is always in very close proximity to the surface. We are not able to assess this constant surface temperature (on a diurnal time scale) with the current observational data.

Lines 21-25 on p.8: Please explain in more detail the asymmetrical retention pattern and the consequences of this.

Corrected by changing the mentioned paragraph.

Line 10 on p. 9: Again, GC-Net firn temperatures can be used here.

Please see answer to question 2 (major points).

Lines 6-7 on p. 10: How good is this threshold? Where does it come from? Please more support for this is needed.

This threshold is derived from remote sensing of firn aquifers. It is estimated to be the approximate sensitivity of the accumulation radar to detect liquid water in the firn (Miège et al., 2016). We added the sentence:
The above-mentioned threshold for firn aquifer delineation is based on a sensitivity estimation of the NASA Operation IceBridge accumulation radar to detect liquid water in firn (Miège et al., 2016).

**Referee 2 (Max Stevens)**

Summary:
This paper presents results of an investigation of the liquid water balance (LWB) of the Greenland Ice Sheet using firn/snow-model (SNOWPACK). The authors' goals are to quantify the components of the LWB spatially and how those quantities have changed in recent decades; to investigate temporal and spatial patterns in refreezing and how those affect the firn; and to assess the models' ability to simulate firn aquifers. The authors force their model using climatic data from the regional climate model RACMO.

Observations of LWB components are unfortunately scarce, but the authors use available data (GRACE data, firn temperatures, firn-aquifer extents) to evaluate their model performance. The model results compare to GRACE data very well. The model does not reproduce firn temperatures as well, but the results do still compare to the data favourably.

General Comments:
The paper is written and organized well and is a topic of wide interest to the cryospheric-science community. Understanding the LWB is important, as the authors identify, because uncertainty in the runoff component of surface mass balance is a large contributor to sea-level-change estimates. Accurate model simulations are an essential contribution to this scientific issue. Additionally, the authors do a good job of discussing potential sources of model error. I recommend this manuscript for publication with minor revisions.

Thank you very much for this positive and encouraging feedback.

**General points to address:**

The authors mention firn "structure" numerous times. I think to many in the firn community "firn structure" refers to microstructural properties such as grain size, coordination number, etc. In this case, the authors refer specifically to firn temperature and porosity. It may be appropriate to call them by name specifically or use "firn physical properties" as the terminology.

We agree and changed this term throughout the manuscript to "firn properties".

The authors ignore any lateral flow and also any heterogeneous flow (i.e. piping). I would like a bit more discussion on how those might affect the results, or if it is even possible to know at this point. Section 2.1 asserts that pore space downhill is often filled, but can enough hydraulic head be generated to drive a significant amount of flow? Is there enough data on piping available to do an easy scale analysis of how much heat could be delivered (how deep, how fast?)

The reviewer is right that these two potentially important processes (lateral flow/heterogeneous percolation) are not considered in the current framework. The main reason for not including these processes in our study is the lack of observational data to constrain the simulation on a GrIS-wide scale. This renders it impossible to quantify the magnitude of these processes – even in a simple scale analysis. Subsequently, we listed some more detailed consideration about the two processes:

- Lateral water transport: It is likely that some fraction of the lateral runoff is retained along its path to the ocean in various parts of the glacier system (e.g. accumulation in subglacial lakes, refreezing in cold snow/firn or storage in firn aquifers). However, the limited observational data make it impossible to quantify the magnitude of this process on a GrIS-wide scale. We can only speculate that, at least in terms of near-surface water retention, the magnitude is rather small due to the mentioned seasonal upward migration of the melt area and the associated depletion of pore space and cold content in lower elevated areas.
- Heterogeneous percolation: There are different methods available to model this process, like the statistical approach of Marchenko et al. (2017) and the physical-based approach of Wever et al. (2016). However, both methods contain at least two tuning parameters that require observational data to calibrate the models. In addition, the approach by Wever et al. (2016) is computationally expensive (it demands a high vertical resolution and solving the Richards equation twice) which makes it unsuitable for distributed model runs with a large number of grid cells.

We modified Sect. 2.2 to clearly state that we neglect these two processes in our model:

We do not consider heterogeneous percolation (Wever et al., 2016; Marchenko et al., 2017) in our simulation due to an insufficient spatial coverage of observational data to calibrate such routines for the entire ice sheet and/or the too expensive computational demand. […] Lateral flow of runoff is also not considered in our simulation.

In Sect. 3.2, we briefly discuss a location where modelled firn temperature is likely bias due to the neglect of heterogeneous percolation. The sentence about the effect of neglecting lateral flow on horizontal redistribution of mass and energy was shifted from Sect. 2.1 to Sect. 5 (and slightly modified):

This is likely a less relevant issue for horizontal near-surface redistribution of mass and energy, as surface melt typically reaches higher elevated areas later in the season, which means that lower areas are already depleted of pore space and/or cold content and thus do not provide any more storage volume for upstream runoff.

Section 3.2 and figure 4 compares model results to observations. The authors identify biases in the model and discuss lower elevation sites, but what about the higher elevation sites? The model shows a somewhat uniform temperature increase over the period, but the observations are not as spatially coherent, e.g. sites 4-050 and 4-000. Also, the model gets the lower-elevation structure correct for the modern, but does not as well for 1960. Why is this? Does your assumption that 1952/55 would be the same as 1960 break down (I do think that is a very reasonable assumption, however.)

We added the following sentences to discuss the incoherency in observed firn temperature change at higher elevations:

The spatially incoherent firn temperature change (between B 4-225 and B 4-000) in the observations is not reproduced by SNOWPACK, which simulates a uniform temperature increase of ~0.3°. This incoherency in the observations may be partly explained by uncertainties in the measurements caused by errors in the sensor calibration and uncertainties in the applied correction used to retrieve 10 m firn temperature from shallower measurements (Polashenski et al., 2014).

We are uncertain to which part of the transect the reviewer is referring with "lower-elevation structure". Discrepancies between modelled and observed temperatures could again be caused by uncertainties in the temperature measurements (as mentioned above).

Section 3.2 (end): If the different parameterization for fresh-snow density works better, why did you not just use that one? How does this uncertainty affect the results for the higher elevation sites? When you say, "An improved fresh snow density parameterization seems therefore essential to address this inaccuracy", do you mean an entirely new parameterization is needed, or just a new-to-your-model parameterization? Does the Langen (2017) parameterization fit the criteria of an improved parameterization?

Because of computational constrains we could only test the parameterisation by Langen et al. (2017) for the mentioned regions in western and southeastern Greenland. For these areas with rather warm climate conditions, the parameterisation outperforms the one we apply from Kuipers Munneke et al. (2015). For colder conditions however, the parameterisation by Kuipers Munneke et al. (2015) is performing better and the parameterisation by Langen et al. (2017) yields too low initial densities for SNOWPACK.

A main issue in our opinion is the proper disentanglement of processes that influence near-surface snow density:

- Density of freshly fallen snow
- Density of deposited snow from snow drift → particles become smaller due to collisions/enhanced sublimation (Groot Zwaaftink et al., 2013)
- Vapour fluxes
- Melt and refreezing

With available fresh snow density parameterisations, it's often unclear which of these processes are considered in the parameterisation and which have to be explicitly modelled. It would be necessary to test SNOWPACK with different available (or newly derived) parameterisations that optimally depend on meteorological parameters, in contrast to the parameterisation by Langen et al. (2017), and compare simulated densities with snow/firn density from locations with various climate conditions to end up with a significantly improved fresh snow parameterisation. Developing such a new expression is clearly beyond the scope of this manuscript.

We simply replaced the last sentence of this paragraph to state our intention more clearly:

This model inaccuracy should be addressed in the future by testing available or newly derived fresh snow density parameterisations with SNOWPACK for various climate conditions on the GrIS.

Section 4.1: Please clarify: You say "changes in the retained liquid mass (dM_ret/dt) are even smaller...". Equation 1 defines dM_ret/dt as the liquid water balance. Is retained liquid mass the same as the LWB? In that sentence, does the quantity in parentheses (dM_ret/dt) refer to 'changes in retained liquid mass' or to 'retained liquid mass'? I don't doubt the science here but it was confusing to read. If changes in LWB, defined as dM_ret/dt, are indeed small, then it might imply that is it not an important term in changes in SMB.

The liquid water balance, as stated in Eq. (1), represents the sum of all liquid water mass fluxes through the upper and lower boundaries of the snow/firn column and refreezing, which is an internal sink for liquid water. The residual of this balance is $dM_{ret}/dt$, the change of retained liquid mass in the snow/firn column with time. Horizontally integrated values of $M_{ret}$ (e.g. over basins) are comparably small, hence values of $dM_{ret}/dt$ are also small. $M_{ret}$ can be substantial for model grid cells with firn aquifers (see Fig. 7 in Steger et al. 2017) but changes in the retained liquid mass ($dM_{ret}/dt$) are typically still small compared to components on the right-hand side of equation (1). We rephrased the following sentence in the manuscript to clarify this:
Changes in the retained liquid mass ($dM_{ret}/dt$) are even smaller than components on the right-hand side of Eq. (1), particularly when integrated over basins, and are thus not presented.

End of page 8, start of page 9, and Figure 8: Please clarify your language: Basin 4 shows a very large increase in firn temperature (due to refreezing I believe), but then you talk about how changes in t_skin are also important (but basin 4 does not show increase in t_skin). Adding a few sentences to clarify this would help – which phenomenon is important where?

There are two important processes that influence local firn temperature:
- Warming/cooling by heat conduction (e.g. caused by a change in surface temperature)
- Warming by release of latent heat from refreezing
In the addressed paragraph, we intended to state that for Basin 4, refreezing is clearly the dominant factor for the increase in firn temperature, whereas for Basin 2, the increase in firn temperature is rather caused by an increase in the surface temperature. We reformulated this paragraph to clarify:
In contrast to other basins, Basin 2 reveals a relatively constant firn temperature increase at lower elevations. This increase is not only caused by the rather small increase in refreezing and the associated latent heat release, but also by an enhanced vertical heat flux from the surface through an increase in surface temperature (Fig. 8d). Surface temperatures changes show a distinct spatial variability, with the largest increases occurring in the northeastern part of the ice sheet, where temperature increases by more than 1.5° C.

**Specific/technical corrections:**

Use of units throughout: in some places, the authors use units of kg/m^2/a (e.g. Figure 5) and in others w.e./a (e.g. Figure 6). It would be good to have consistency throughout. I slightly prefer m w.e./a in this context because it is slightly more intuitive.

Corrected by consistently using m w.e. (or mm w.e. for smaller values) in all figures and in the text.

Use of vague language throughout: several instances of "it seems" or "apparently". Just say what you mean directly. E.g. page 5 line 19: change to "it is reasonable".

Corrected by rewriting sentences with "it seems" (Sect. 1, 3.2, 5) and "apparently" (Sect. 3.2, 4.3)

Page 1, Line 5: "good model performance" is vague; perhaps "indicate good model- observation agreement" or something along those lines

Corrected.

P1L7: "increases with" change to "increases at"

Corrected.

P1L13: be aware that upward could also mean forming at a shallower depth "migration of firn aquifers to higher elevations"

Corrected.

P1L20-21: put the e.g. section in parentheses to break up the sentence more clearly; change to "and the darkening"

Corrected.

P1L24: change to "suggest that modelled refreezing"

Corrected.

P3L15: perhaps a new paragraph at "The Greenland mass..."

Corrected.

P4L1: please clarify: do you mean densification scheme, as in how density changes with time, or the parameterization for new snow density that you use?

Thank you for pointing out this error. We indeed meant the fresh snow density parameterisation and not snow densification. We change the sentence to:
The enhanced near-surface snow compaction due to strong winds, which is implemented in SNOWPACK for Antarctic simulations (Groot Zwaaftink et al., 2013), is switched off, because the applied fresh snow density parameterisation already accounts for this effect.

P4L9: get rid of word Furthermore.

Corrected.

P4L19: it is a bit unclear what you mean by "indirectly". "over" is probably not the best word here.

We rephrased this sentence:
Due to the lack of direct refreezing observations, we assess the model's performance in terms of refreezing indirectly by comparing the spatially integrated SMB and local snow/firn temperatures to observations.

P4L25: start new paragraph at "A comparison", rather than at line 28.

Corrected.

P5L3: do you mean delaying runoff by 18 days in the model? Please clarify.

We rephrased this sentence:
Van Angelen et al. (2014) demonstrated that the monthly error between detrended modelled SMB and GRACE on a GrIS-wide scale could be minimised by delaying simulated runoff by 18 days.

P5L7: mean seasonal amplitude in what?

Clarified by changing the sentence to:
The mean seasonal amplitude of the detrended modelled MB derives by ~30 % from winter accumulation and summer melting of seasonal snow over the tundra (Fig. 3b).

P5L7: change to "A too-early modelled snow..."

Corrected.

P5L13-14: remove semi-colon and e.g., change to ". . .et al. 2015), or accumulating..."

Corrected by rewriting the sentence to:
Finally, runoff may also be retained in the hydrological system of the tundra by refreezing in soil, ponding on frozen ground (Johansson et al., 2015), accumulating in surface lakes (Mielko and Woo, 2006) and storage in terrestrial aquifers.

P5L20-21: change to "Figure 4 shows that SNOWPACK..."

Corrected.

P6L5: I think you have not defined IMAU-FDM acronym prior to this use.

Corrected.

P6L8: The sentence starting "Due to a different..." is a bit awkward to read – try to rephrase to be active voice.

Corrected by rewriting the sentence to:
Compared to IMAU-FDM, the overestimation of this zone is less pronounced in SNOWPACK owing to a different densification scheme, which is more accurate for relatively warm conditions (Steger et al., 2017).

P7L12: 47% refreezes – how has this changed since 1960?

The reviewer is kindly referred to Fig. 6 and Table 1. Figure 6 (panel for GrIS) shows that the refreezing fraction is rather constant for the first period (1960 – 1989) with some interannual variability. The fraction decreases during the second period (1990 – 2014) according to the linear trend stated in Table 1.

P7L17: list the four most relevant components in parentheses.

Corrected.

P7L20: "Remarkably" – why is this remarkable? Would you have expected other regions to also have more rain? Is there more precipitation in total in that region, or higher rainfall as a percentage of total precipitation – how does that compare to other areas?

We expected significant increases in rainfall also in other regions, particularly in the more southerly located basins. We checked time series and linear trends (for 1990–2014) for rainfall, snowfall, total precipitation and rainfall fractions for the GrIS and the eight basins. The only significant trends that were found are from Basin 8, where both rainfall and rainfall fraction exhibit a positive significant trend. We included this finding also in the manuscript:
This increase is not caused by a change in total precipitation but by a significant increase in the rainfall fraction in this area.

P7L27: change "does" to "do"

Corrected.

P7L31: change "one" to "trend"

The sentence was modified.

P8L6: Is melt climate? I might suggest that temperature is climate, and melt is a result (but maybe I am wrong or nit-picking or both)

We agree with the reviewer's comment that melt is not really a climatic factor but rather the result of the climatic factor temperature. Hence, we change the sentence as follows:
Refreezing is a process that strongly depends on local climate, i.e. particularly on surface temperature which is the main driver for melt, and therewith on seasonality and elevation (Fig. 7).

P8L13: do you mean "the seasonal decrease in refreezing at higher elevations is caused by . . ."?

We merged this sentence with the previous one to clarify our statement:
Therefore, refreezing at lower elevations persists throughout the melt season but with lower rates than in spring due to a gradual decrease in the firn cold content.

P8L23-26: Are these sentences about hysteresis needed? They seem distracting and not relevant to me, but if they are relevant, include some discussion about how it affects your results and conclusions.

We decided to rewrite this paragraph and remove the term hysteresis, because it would require too much additional explanation.

P8L29: get rid of word "even"

Corrected.

P8L30: change to "causes melt and refreezing..."

Corrected.

P9L18: change "higher" to "larger"

Corrected.

P9L18 and Figure 10: perhaps adding a panel to figure 10 to show pore space across the transect?

We agree that pore space would be an interesting additional parameter to show. However, because it can be inferred from firn density and to keep the figure size reasonable, we decided not to include this additional parameter.

P10L10: get rid of word "apparently"

Corrected.

P10L16: get rid of apparently; say something like, "Instead, in these regions liquid water drains into..."

Corrected.

P10L20: Change to: "Comparing the modelled depth of the firn-aquifer top to observations is difficult..."

Corrected.

P10L23: change to "unsaturated wet layer"

Corrected.

P10L26: unclear what you mean by advection of cold interior ice here – please expand on the physics of what is going on.

What we meant is the following: We initialise firn temperatures for our simulation with a vertically constant value equal to the RACMO2.3 skin temperature (1960 – 1979 average). Next, we correct this temperature for latent heat release by refreezing with a parameterisation (Steger et al. 2017). As a result, firn temperatures are mostly initialised with temperatures close to 0°C at firn aquifer locations. With the assumption of a zero-heat flux at the base of the model domain, no heat sinks are present and modelled deeper firn temperatures remain at 0°C. The base of modelled aquifers is then determined by firn compaction (liquid water is "squeezed" out if there is no pore space left). Recent observations indicate however that refreezing conditions often prevail at the base of aquifers (and that

this effect thus limits the vertical extent of aquifers). Apparently, there is a "cold reservoir" beneath aquifers that induces a downward directed heat flux. This colder ice is advected laterally from the interior of the ice sheet, because it originates from higher elevations.
The sentence was however removed and we generally modified this section due to overlaps with Steger et al. (2017). Wherever necessary, we refer to this publication in the manuscript.

P10L28: would initializing the model with lower temperatures be a physical thing do to? Is there any reason to believe that firn temperatures were lower in 1960 (assuming that is the start of the simulation)?

Initialising deeper firn temperatures with lower values and applying a downward directed heat flux at the lower boundary of the model seems reasonable in view of recent observations that reveal refreezing conditions at the base of firn aquifer, see the explanation above. However, a more detailed knowledge of the thermodynamic conditions deeper in the firn is required to apply such modifications.

P10L32: Specify the time period over which that expansion is occurring.

We changed this sentence to clearly state the different time periods for modelled and observed firn aquifer expansion:
Figure 11 shows an expansion of the firn aquifer to higher elevations during the simulation period (1960–2014). This trend is in line with observations (2010–2016), which indicate an inland expansion of aquifers in this region (Miège et al., 2016; Montgomery et al., 2017).
Evaluations of radar measurements for an even longer period (1993 – 2016) confirm the inland expansion of firn aquifers – at least for the Helheim region (Miège et al., in preparation).

P11L4-5: are these trends or just changes?

We changed this sentence and the one before to clarify our statements:
In Basins 4 and 5, the mean surface elevation at which firn aquifers are modelled rises by ~200 m during the simulation period (1960 – 2014). Upward migration of firn aquifers is also apparent in other basins, where Basin 3 and 6 reveal a smaller change of 125 and 90 m, respectively, and Basin 8 a larger elevation increase of 215 m.

P11L11: hyphen in ice-sheet margin

Sentence was removed due to overlap with Steger et al. (2017).

P11L12: s in aquifers; change "i.e." to "specifically"

Corrected by adding an "s" to "aquifer". However, we think that "i.e." fits better in this context than "specifically" – hence we did not amend this expression.

P11L14: no comma after simulation

Corrected.

P11L16: get rid of word "the" before aquifer formation

Corrected.

P11L26: get rid of word "intuitively"

Corrected.

P13, end: also mention piping perhaps

As Anonymous Referee #1 correctly stated, there was some thematic overlap between this manuscript and Steger et al. (2017). We removed this overlap and are thus not mentioning heterogeneous percolation (piping) in the conclusion but refer to Steger et al. (2017), where this process is already suggested in the conclusions as a potential model improvement.

Figure 3 caption: change to "shaded areas illustrate"

The shaded areas (interannual variability) were removed from the figure because they are not discussed in the main text.

Figure 5: It might be nice to have basin numbers labelled or on this figure to avoid flipping to figure 2. Why is refreezing the only parameter you show fraction for? You could, for each LWB component, show the value (as you do) with the fraction of total in parentheses. So, for example, basin 1 in panel e would have the value 14 (30%). That way the reader could see the fraction for all of the components. Also on this figure: The outlines of the basins are tough to see (make them darker and make the ELA a dashed dark line?). The colour scales make it tough to see what is going on near the margins – not sure how to fix that.

Basin numbers are now provided in panel (a). As the reviewer suggests, it would indeed be possible to show other parameters than refreezing as fractions (e.g. melt as a fraction of the total liquid water input). However, as we are focusing on liquid water retention in snow/firn, we think that showing fractions for other values than refreezing would distract from the focus.
To enhance visibility of the basin's boundaries, we increased the line width and selected a darker grey for the lines. We agree with the reviewer that the colour scale makes it difficult to recognise patterns near the margins. This issue is caused by the fact that most plotted parameters exhibit the highest gradients close to the margins. The use of a non-linear colour scale would improve readability but it would also distort the presented values. Therefore, we decided to keep the current colour scale.

Figure 6: the legend could be larger at the bottom, especially with the thickness of the lines

Corrected (Fig. 14 was improved likewise).

Figure 8: Perhaps move the x-labels to the bottom so that the numbers and labels are next to one another. It is a bit confusing – is this showing the change in the average values for 1960-1989 subtracted from the average values for 1990 – 2014? Also on this figure: legend could be larger

We shifted the units to the bottom of the panels and increased the legend according to the reviewer's suggestion. The figure caption was also adapted to clarify the description of the presented data: Temporal changes in refreezing, firn air content, firn temperature (averaged over 2–10 m depth) and skin temperature in 100 m elevation bins. The difference shows the 1990–2014 average minus the 1960–1989 average.

Figure 10: An additional panel showing the elevation/surface profile of the ice sheet could be useful here.

We refer the reviewer to the axis at the bottom of Fig. 10, which shows the surface elevation of the ice sheet. We agree that an additional panel with a plot of the surface elevation would enhance readability but we omit this panel to keep the vertical size of the figure in a reasonable range.

Figure 11: inset panels are quite small.

We enlarged the inset panels in (b) and (c), increased the line width and made all fonts bold to enhance readability.

Figure 13: Put a label along the colour bar: Firn Air Content (m). The dots here make this a bit tough to understand. Perhaps making the aquifer dots more contrasting colours? Or use crosses? The grey dots are tough to see.

We added a label along the colour bar as suggested by the reviewer. Additionally, we changed the colours of the dots to increase readability. We also changed the colour bar for firn air content to grey-scale, because this quantity is a rather secondary information in this figure (compared to the occurrence of firn aquifers).

Table 1: The units are difficult to understand here. Does m w.e. aˆ-1 (25 a)ˆ-1 refer to melt, rainfall, runoff, and refreezing? Since you state in the caption that these trends are 1990 – 2014, does it not suffice to say the trend is in m w.e. per year?

We agree with the reviewer that the units are somewhat difficult to read. Therefore, we converted the numerical values to units that are more intuitive:
- Trends in mass fluxes: (mm w.e. $a^{-1}$) $a^{-1}$
- Trend in refreezing fraction: % $a^{-1}$

**Additional references**

Mottram, R., Boberg, F., Langen, P., Yang, S., Rodehacke, C., Christensen, J. H., and Madsen, M. S.: Surface mass balance of the Greenland ice sheet in the regional climate model HIRHAM5: Present state and future prospects, Low Temperature Science, 105:115, doi: 10.14943/lowtemsci.75.105, 2017.

Schmidt, L. S., Aðalgeirsdóttir, G., Guðmundsson, S., Langen, P. L., Pálsson, F., Mottram, R., Gascoin, S., and Björnsson, H.: The importance of accurate glacier albedo for estimates of surface mass balance on Vatnajökull: evaluating the surface energy budget in a regional climate model with automatic weather station observations, The Cryosphere, 11, 1665-1684, doi: 10.5194/tc-11-1665-2017, 2017.

[revised manuscript text omitted]

---

## Author Response (AR2)

Dear editor,

Thank you very much for the further comments to improve the manuscript. Please find below our responses to the individual points. To facilitate readability, responses are in blue and modifications in the manuscript in red.

All minor suggestions regarding grammar, style, block format and references are considered and marked in red in the attached manuscript version. Regarding the major points:

Page 1: *Indirect evaluations is confusing. Please state more clearly what you mean here.*

We agree with the editor that the word "indirect" is confusing without further explanation. Instead of explain it more clearly in the abstract (where the explanation would required to much space), we removed it from this sentence. The concept of indirect evaluation is subsequently explained in Sect. 3.

Page 2: *The authors have neglected to mention the mechanism of buried lakes in the western GRIS as a storage mechanism that is also neglected in fig1. You should mention in the text or clarify why they are not important for this type of modeling study.*
*https://www.the-cryosphere.net/9/1333/2015/tc-9-1333-2015.pdf*

We mention this feature in the revised manuscript by rephrasing the sentence:
The amount of water stored in supra-glacial lakes, which may be buried with snow in winter and hence retain liquid water perennially, is thereby rather small compared to the magnitude of supra-glacial river fluxes (Smith et al., 2015; Koenig et al., 2015).

Page 2: *Define upper part i.e. (~50-100 m)*

We added an approximate range for the upper part of the ice sheet.

Page 7: *New paragraph here I think. This sentence and the previous to not flow. Please reread and correct for clarity.*

The section "At higher elevations in western Greenland..." belongs to the previous part, as it states where SNOWPACK simulates a pronounced warming (in contrast to the lower elevations where the warming is not reproduced due to a firn density overestimation). We also think that the flow is not interrupted by this sentence (and do not see a distinctively better way to arrange the text) – therefore we did not change this section.

Page 11: *Remove last sentence of section 4.2*

Corrected.

Page 11: *Miège et al. (2016) for 2010 to 2014 mapping.*

Corrected. For the observed horizontal firn aquifer extent in Fig. 11, we use now data from Miège et al. (2016) by intersecting our transect with the derived firn aquifer polygons. This observational data indicates a larger horizontal extent of the aquifer compared to the previously used data (Montgomery et al., 2017). We also adapted the reference in the caption for Fig. 11.

Page 11: *reference here should also be to Miège who has done all the aerial mapping of the aquifer.*

This sentence was removed from the manuscript. The section was rephrased to clarify the intended statement:
At lower elevations, SNOWPACK simulates a larger horizontal extent of the aquifer than inferred from radar data (Fig. 10 and 11). Apart from model uncertainties, this disagreement may be caused by different criteria used to delineate firn aquifers, where radar-derived mapping relies on the detection of a water table. The realisation of such a table may be prevented in this area by drainage of the aquifer into crevasses, where water either refreezes or enters the subglacial drainage system (Poinar et al., 2017).

Page 12: *Montgomery et al. are using seismic and radar for the top depth of the aquifer. Which are you using? Please clarify.*
*The seismic do have a more gradual transition than the radar data.*

We compare our simulation results to the average depth of the water table stated in Montgomery et al. (2017). As we understand from this study, the base of the aquifer was inferred from seismic measurements whereas the depth of the water table was derived from radar data. We therefore did not change or extend the sentence in question.

Page 12: *Please add an additional sentence here explaining that the Miège et al paper provides a minimum extent of the aquifer due to the inconsistent flight patterns of airborne data.*

Corrected by adding the sentence:
The observational derived estimate provides a minimum extent of the aquifer due to inconsistent flight patterns of airborne data.

Page 14: *Weak ending. Please add an additional sentence in summary concluding the work you have done.*

We added a final sentence to strengthen the ending of the manuscript:
Nonetheless, the modelled MB and firn temperatures presented in this study compare favourably with remote sensing and in situ data and allow for a detailed evaluation of the GrIS's LWB between 1960–2014.

[revised manuscript text omitted]
 by in situ observations in 2011 (Forster et al., 2014) and mapped from radar measurements for 2010–2014 (Miège et al., 2016). The grey shaded area in Fig. 11 indicates the horizontal extent of these mapped aquifers. The combination of RACMO2.3 and SNOWPACK underestimates the the upper limit of the firn aquifer's horizontal extent by approximately 50 m in elevation if one assumes only small changes in aquifer extent between 2010 and 2014. A brief sensitivity test of SNOWPACK with a lower

25   fresh snow density, as described in Sect. 3.2, yields a firn aquifer that reaches higher elevations and thus reduces the mismatch. The reason for this improvement is that the lower near-surface firn density reduces the conductive heat loss of the aquifer to the atmosphere in winter. At lower elevations, SNOWPACK simulates a larger horizontal extent of the aquifer than inferred from radar data (Fig. 10 and 11). Apart from model uncertainties, this disagreement may be caused by different criteria used to delineate firn aquifers, where radar-derived mapping relies on the detection of a water table. The realisation of such a table

[revised manuscript text omitted]